# Modelling associated factors of maternal age at first birth in Ethiopia: Gamma regression approach

**Adimias Wendimagegn Agegnehu**[1]*, **Butte Gotu Arero**[2]

**1** Department of Mathematics, Kotebe University of Education, Addis Ababa, Ethiopia, **2** Department of Statistics, Addis Ababa University, Addis Ababa, Ethiopia

* adimaswend@gmail.com

## Abstract

### Background

Maternal age at first birth (AFB) is a key indicator of reproductive health, influencing maternal and child outcomes, population dynamics, and socio-economic well-being. Early childbearing increases risks of maternal morbidity, poor neonatal outcomes, and intergenerational poverty. Despite progress in Ethiopia, disparities in AFB persist across regions, religions, and residential settings, particularly among women who have already given birth. This study determines the magnitude and direction of effects from religious, regional and residential determinants of maternal AFB in Ethiopia.

### Methods

Data were drawn from 5,839 women included in the 2019 Ethiopia Mini Demographic and Health Survey (Mini EDHS 2019). A comparative modeling framework was employed; fitting Generalized Linear Models (GLM), Generalized Linear Mixed Models (GLMM), and distributional Gamma regression models to examine the consistency of the coefficients for covariates across different modeling framework. Both classical estimation (Maximum Likelihood Estimation) and Bayesian inference via Markov Chain Monte Carlo (MCMC) were used.

### Results

Among women in Ethiopia who had experienced a first birth by the time of the survey, the average maternal age at first birth was 18.7 years(95% CI: [18.2, 19.3]), indicating that a substantial proportion of women begin childbearing at the lower limit recommended by the World Health Organization. Regional differences in mean maternal age at first birth were notable: women in Amhara (4%), Oromia (3%), SNNPR (3%), Benishangul-Gumuz (3%), and Gambela (8%) had earlier average maternal age at

**Data availability statement:** All relevant data are within the manuscript and its Supporting information files.

**Funding:** The author(s) received no specific funding for this work.

**Competing interests:** The authors have declared that no competing interests exist.

**Abbreviations:** ADHS, Angola Demography and health survey; AFB, Age at first birth; AIC, Akaike Information Criterian; CSA, Central Statistical Agency; EA, Enumeration Area; EDHS, Ethiopian Demography and Health Survey; EPHI, Ethiopian Public Health Institute; ESS ($n_{ff}$), Effective Posterior Sample size; EU, European Union; GLM, Generalized Linear Model; GLMM, Generalized Linear Mixed Model; HMC, Hamiltonian Monte Carlo; ICF, International Classification of Functioning; Liglik, Log likelihood; LOO, Leave-One-Out; MCMC, Monte Carlo Markov Chain; MCSE, Monte Carlo Standard Errror; MLE, Maximum Likelihood Estimation; NUTS, No-U-Turn Sampler; PPC, Posterior Predictive Checking; QQ, Quantile-Quantile; SNNPR, South Nation Nationality and People Region; UNICEF, United Nation International Children's Emergency Fund; WHO, World Health Organization.

first births, whereas those in Addis Ababa (11%), Somali (4%), and Dire Dawa (4%) delayed childbirth compared to Togray. Variability in AFB also differed by region, with higher dispersion observed in Afar (37.7%), Benishangul-Gumuz (37.7%), Somali (36%), Gambela (32.3%), SNNPR (27.1%), and Harari (25.9%), while Addis Ababa (22.1%) and Dire Dawa (13.1%) showed lower variability. Women who have already given birth and identify as Orthodox or Protestant tend to experience a higher average age at first birth, with increases of approximately 4% and 3%, respectively, relative to Muslim women. Conversely, women classified under the "other religions" category tend to have their first child approximately 5% earlier than Muslim women. Residence influenced both mean and variability: rural women who had already given birth experienced first births about 5% earlier than urban women and exhibited 7.3% greater dispersion, indicating more heterogeneous reproductive patterns.

## Conclusion

Maternal age at first birth in Ethiopia remains low (18.7 years), with significant disparities in mean and variabilities across religion, residence, and region. Women from rural areas, Muslim backgrounds, and regions such as Amhara, Oromia, Benishangul-Gumuz, SNNPR, and Gambela are more likely to give birth earlier, while urban women, particularly in Addis Ababa, Dire-Dawa, and Somali, tend to delay childbearing. The persistence of very early childbearing (as young as 10 years) highlights urgent reproductive health challenges. These findings call for context-specific policies: expanding girls' education, delaying marriage, and strengthening youth-friendly reproductive health services in rural and high-risk regions; engaging religious leaders and institutions as partners in promoting delayed childbearing; and addressing structural inequities through regionally tailored programs that respect cultural and pastoralist lifestyles.

## Introduction

The timing of a woman's first birth is a critical determinant of maternal and child health, fertility trajectories, and socio-economic outcomes [1,2]. Early childbearing, particularly in settings with limited access to contraception, is associated with higher lifetime fertility, interrupted education, reduced career opportunities, economic insecurity, and increased risks to maternal and child health [2,3]. Adolescent mothers, especially those under 18, face substantially higher maternal morbidity and mortality: girls aged 10–14 are five times, and those aged 15–19 twice, as likely to die from pregnancy- or childbirth-related complications compared with women aged 20–24 [3–5]. They are also more susceptible to obstetric complications such as eclampsia, fistula, systemic infections, and long-term chronic illnesses [6–9]. At the population level, early motherhood contributes to high fertility rates and rapid population growth, particularly in regions with low contraceptive prevalence [10]. Children born to mothers under 16 have higher morbidity and

under-five mortality risks [11]. Consequently, the World Health Organization recommends delaying first childbirth until at least age 18 [12]. Conversely, delaying first birth beyond age 35 carries elevated risks of miscarriage, multiple births, chromosomal abnormalities, gestational diabetes, and maternal mortality [3], highlighting the importance of identifying an optimal timing for first birth.

Globally, early marriage and adolescent pregnancy remain major challenges in low- and middle-income countries. Over 30% of women marry before 18, and 14% before 15 [13], with 95% of births to women aged 15–19 occurring in these settings [13]. Sub-Saharan Africa exhibits the highest fertility rates and low contraceptive use [14–16]. Median age at first birth varies widely: 26.3–31.3 years in the European Union [17], 20.2 years in East Asia and the Pacific [18], and 19 years in Sub-Saharan Africa, with national variation such as 20 in Nigeria, 19 in Ghana, 16.34 in Bangladesh, and 19.2 in Uganda [19,20]. In Angola, 5% of girls aged 15–19 give birth before 15 [21]. Across 36 countries, children of mothers under 18 face a 46% higher risk of dying before age five, while for mothers aged 18–19, the risk is 12% higher than for mothers aged 20–34 [22].

Ethiopia faces significant early childbearing challenges amid high population growth (2.57% per year) and an average fertility of five children per woman [23]. In 2021, 39.56% of the population was under 14, and 21.1% were aged 15–24 [24]. Among women aged 15–19, 13% were pregnant or mothers, and 22.2% of women aged 20–24 had given birth before 18 [24,25]. Ethiopia ranks fourth globally in girls married before 18 [26], and over 76% of married women aged 15–19 had never used contraception [25]. Median age at first birth (AFB) was 19 in 2005, rising slightly to 19.2 in 2011 and 2016, then declining to 18.7 in 2019; roughly half of women aged 25–49 had their first child before 20 [25,27,28]. Key determinants of early AFB include early sexual debut, low educational attainment, early marriage, large spousal age gaps, rural residence, sociocultural norms, contraceptive non-use, employment, autonomy, and religion [12,29,30]. Policy efforts in Ethiopia aim to reduce early pregnancies from 13% to 7% and increase the average age at first marriage from 17 to 18 by 2025 through female education and economic empowerment [31]. Despite these interventions, early childbearing remains prevalent, highlighting the need for evidence-based research.

Most prior studies on AFB use survival analysis, particularly Cox proportional hazards models, which assume proportional hazards and may inadequately capture variability. Parametric models such as the Weibull accelerated failure time model have also been applied, but often fail to account for covariate effects on dispersion [31]. Additionally, many analyses include variables measured at survey time rather than at first birth, limiting causal interpretation and precision. Skewness and heterogeneity in AFB data are frequently overlooked, potentially biasing estimates and obscuring relationships. This study addresses these gaps by focusing exclusively on women who have already given birth, avoiding censored observations.

Understanding maternal age at first birth requires situating individual reproductive behavior within broader social, cultural, and economic contexts. In Ethiopia, factors such as region, religion, and place of residence are intertwined with norms, opportunities, and constraints shaping the timing of childbearing. Regional variation may arise from differences in economic development, infrastructure, and service provision [29]. Regions with higher urbanization and economic growth often provide greater access to education, healthcare, and employment, which can delay marriage and first birth. Conversely, in less developed regions, limited schooling opportunities, reliance on agrarian livelihoods and persistent traditional norms may encourage earlier entry into marriage and childbearing.

Religious affiliation can influence first birth timing through doctrinal teachings, community norms, and institutional roles in shaping gender expectations [30]. These effects may differ by place of residence; for example, the same religion may manifest differently in urban versus rural contexts. Urban women generally have better access to education, formal employment, and modern contraceptives, which can postpone first birth [25,26]. Rural women may face early school dropout, agricultural labor demands, and stronger enforcement of traditional marital norms, resulting in earlier first births. These predictors can also interact for instance, the delaying effect of higher education may be stronger in urban areas, or religious influences may be amplified in rural settings. Variables excluded from the empirical model due to temporal

inconsistency, such as current educational attainment and marital status, may act as mediators linking region, religion, and residence to reproductive timing.

Using a Gamma regression framework under generalized linear models (GLM), generalized linear mixed models (GLMM), and distributional Gamma regression, this study models both the mean separately and jointly with variability of AFB. The study aims to quantify the influence of geographical, religious, and residential factors on both the average and dispersion of maternal age at first birth in Ethiopia, while evaluating the adequacy of gamma regression models under classical and Bayesian estimation approaches. By doing so, it provides robust, context-specific evidence to guide interventions that delay early childbearing and improve maternal and child health outcomes.

## Materials and methods

### Study design

A cross-sectional study was conducted to examine geographical, religious, and residential factors associated with maternal age at first birth (AFB) in Ethiopia.

### Study setting and population

Ethiopia is located in the Horn of Africa and is bordered by Eritrea to the north, Djibouti and Somalia to the east, Kenya to the south, South Sudan to the west, and Sudan to the northwest. Administratively, the country is divided into nine regional states Tigray, Afar, Amhara, Oromia, Somali, Benishangul-Gumuz, Southern Nations, Nationalities and Peoples' Region (SNNPR), Gambela, and Harari and two chartered cities, Addis Ababa and Dire Dawa.

This study utilized secondary data from the 2019 Ethiopia Mini Demographic and Health Survey, which was conducted from March 21 to June 28, 2019. The survey was implemented by the Ethiopian Public Health Institute in collaboration with the Central Statistical Agency (Ethiopia) and the Federal Ministry of Health (Ethiopia), with technical assistance from ICF. The Mini EDHS is a nationally representative household survey designed to provide reliable estimates of demographic and health indicators at the national and regional levels, as well as by place of residence (urban and rural). The dataset used in this study was obtained from the Demographic and Health Surveys Program website after formal approval of a data access request. Ethical clearance for the survey was obtained by the DHS Program from the Institutional Review Board of ICF and the national ethics review committee in Ethiopia [32].

The target population of the Mini EDHS includes women aged 15–49 years residing in selected households across Ethiopia. However, for the purposes of this analysis, the study population was restricted to women aged 15–49 years who had experienced at least one birth before the survey date.

### Sample design

The 2019 Ethiopia Mini Demographic and Health Survey employed a two-stage stratified cluster sampling design. In the first stage, 305 census enumeration areas (EAs) were selected from the national sampling frame, including 93 urban clusters and 212 rural clusters. In the second stage, a systematic sampling method was used to select approximately 30 households from each selected enumeration area, resulting in a total sample of 8,885 households. The sampling design ensured adequate representation across regions and urban–rural areas, with proportional allocation applied to most regions and additional sampling consideration for the three largest regions: Amhara, Oromia, and SNNPR. This design allowed the survey to produce nationally and regionally representative estimates of key demographic and health indicators.

### Analytical sample

Age at first birth is inherently a time-to-event variable, and women who have not yet experienced a first birth represent right-censored observations. In the present study, the analysis was intentionally restricted to women who had already

experienced their first birth by the time of the survey. This restriction aligns with the study objective of modeling the timing of first birth among women who have transitioned to motherhood and allows direct estimation of the conditional mean and dispersion of maternal age at first birth.

Including women who had not yet experienced a first birth would require survival analysis methods to account for censoring, which was beyond the scope of the present modeling framework. Consequently, the analytical sample consisted of 5,839 women aged 15–49 years who reported having had at least one live birth at the time of the survey. Women who had not yet experienced a first birth were excluded from the analysis.

While this restriction allows detailed modeling of the distribution of age at first birth among mothers, it changes the interpretation of results. Therefore, the findings should be interpreted as applying to women who had already experienced first birth at the time of the survey, rather than to all reproductive-age women in Ethiopia.

## Outcome variables

The study outcome variable is maternal age at first child birth.

## Explanatory variables

Region (Tigray, Afar, Amhara, Oromia, Somali, Benishangul, SNNPR, Gambela,Harari, AddisAbaba and Dire Dawa).
Place of residence (Urban, Rural).
Religion (Muslim, Orthodox, Protestant, Catholic, Traditional, Others).

The selection of explanatory variables in this study was guided by considerations of temporal consistency with the outcome variable, maternal age at first birth (AFB). Although maternal education, occupation, and marital status are widely recognized in the literature as important correlates of AFB, these variables in the 2019 Ethiopia Mini Demographic and Health Survey were recorded at the time of the survey rather than at the time when the first birth occurred for most respondents. Because these characteristics may have changed substantially after the first birth event, their inclusion could introduce temporal inconsistency and potential reverse causality. For example, early childbearing may itself influence subsequent educational attainment or employment status, thereby biasing the estimated relationships if these variables were included as predictors.

Nevertheless, the exclusion of such variables may introduce omitted variable bias if they are associated with both the outcome and the retained covariates. For instance, educational attainment is likely correlated with regional differences in access to schooling as well as with delayed childbearing. Consequently, some of the estimated regional differences in maternal age at first birth may partly reflect underlying educational disparities. Similarly, marital status and occupation may be linked to both social norms and reproductive behavior, and their exclusion could lead to over- or underestimation of the associations between the included covariates and age at first birth. These limitations are acknowledged, and the results should therefore be interpreted as associations rather than causal effects.

Given these considerations, the final set of explanatory variables included region, religion, and place of residence. These variables were selected because they represent relatively stable contextual characteristics that are less likely to change following the occurrence of first birth. They capture important geographic, cultural, and social dimensions that may influence reproductive behavior, including differences in access to education and health services, sociocultural norms surrounding marriage and childbearing, and disparities between urban and rural environments. Focusing on these contextual variables allows the analysis to explore patterns and associations in maternal age at first birth among women who had already experienced a first birth at the time of the survey, while minimizing temporal inconsistencies between predictors and the outcome.

## Ethical approval and consent

This study utilized publicly available, de-identified data from the 2019 Ethiopia Mini Demographic and Health Survey (Mini EDHS), obtained with permission from the DHS Program (www.dhsprogram.com). Ethical approval for this secondary

analysis was waived by the Kotebe University of Education Institutional Review Board by letter with ref N̲o̲ Math098/2017, as the data are anonymized and publicly accessible, preventing identification of individual participants. The study adhered to ethical standards for secondary data analysis, with no issues related to dual publication, misconduct, or participant confidentiality.

## Data analysis

The variable of interest in this research is the age at which a mother gives birth for the first time. This is calculated as the number of years from her date of birth to the age at which she delivers her first child. The dataset on age at first birth is considered to be positively skewed and continuous, as it consists of a series of positive values within limited range. When dealing with such data that is continuously distributed, positively skewed, and always positive, one common technique is to perform a log transformation on the data [33]. However, there are other approaches within the generalized linear model framework that may provide a more comprehensive analysis of this type of dataset. It is important to explore these alternative methods to ensure that the analysis of the age at first birth data is both accurate and meaningful. In this study, it was observed that the dataset on the maternal age at first birth obtained from the Mini EDHS 2019 dataset (excluding mothers without birth during survey period) follows a gamma distribution and consists of only positive values.

Gamma regression was chosen because the focus of this study is on modeling the observed age at first birth as a continuous, positively skewed variable, rather than modeling the hazard or timing of the event. The Gamma distribution naturally accommodates skewness and strictly positive values, allowing both the mean and dispersion to be modeled flexibly under GLM, GLMM and distributional Gamma regression frameworks. In contrast, time-to-event models such as the Cox proportional hazards or Accelerated Failure Time (AFT) models are most appropriate when dealing with right censored data or when the objective is to model the hazard rate over time. In this study, all respondents had experienced the event (no right-censoring), and the research objective was to quantify the relationship between regional, religious and residential covariates and the mean age at first birth and variability, making Gamma regression a more direct and interpretable choice.

This study explores Gamma regression models that allow both the mean and shape (dispersion) parameters to depend on covariates. Initially, the mean parameter is modeled within the frameworks of the generalized linear model (GLM) and generalized linear mixed model (GLMM). Subsequently, a distributional Gamma regression model is employed, in which both the mean and variability are regressed on relevant covariates. All models are examined under both classical (maximum likelihood) and Bayesian (MCMC) estimation approaches [33].

## Modeling framework and estimation approaches

The Linear Model (LM) has had a very broad development. LM has developed into the Generalized Linear Model (GLM). Furthermore, the combination of GLM and Linear Mixed Model (LMM) can form a Generalized Linear Mixed Model (GLMM) [34]. The generalized linear model (GLM) is a powerful regression model that merges linear and nonlinear elements, enabling the incorporation of non-normal response distributions. In a GLM, the response variable needs to be a part of the exponential family, which encompasses various familiar distributions like normal, log normal, binomial, exponential, Weibull and gamma distributions. The normal-error linear model is a specific variant of the GLM, making the GLM a versatile tool for empirical modeling and data analysis [34]. Explanatory variables in GLMs can be either continuous or categorical. The GLM determines a separate regression coefficient for each level of categorical variables indicating the need for exercising considerable care in interpreting that coefficient.

The GLM is constructed from three key components:

1. Random Component: This specifies that the conditional distribution of the response variable $Y_i$ (the i$^{th}$ of n independently sampled observations) is determined by the values of the explanatory variables, with the expected

value E $(Y_i)$ = $\mu_i$ for $i = 1,...,n$. It is assumed that $Y_i$ follows a specific probability distribution, such as the gamma distribution. The choice of distribution is contingent upon the type of response variable and the characteristics of the data [35].

2. Systematic Component: This defines a linear predictor, which is a linear combination of the explanatory variables with k predictors, represented as $X_{ik}$. This is expressed as a linear predictor $\eta_i$, where the relationship can be formulated as $\eta_i = \mathbf{X}_i\boldsymbol{\beta}^T$ where $\boldsymbol{\beta}$ being the vector of unknown regression coefficients that need to be estimated [35].

3. Link Function: This component connects the linear predictor to the mean of $Y_i$. The link function is a monotonically increasing function, and the selection of this function depends on the distribution of $Y_i$. For instance, when $Y_i$ represents a positive integer, the mean E $(Y_i)$ = $\mu_i$ must also be non-negative, prompting the use of a log link function, which expresses the mean in terms of the entire real line. In this case, the link function is represented as log $(\mu_i)$ effectively relating $\mu_i$ to the linear predictors. Consequently, log-linearity for the mean is frequently employed as the link function within the regression model.

Generalized linear mixed models (GLMMs) is an extension of GLM where random effects are included in the model. GLMMs are a natural outgrowth of both linear mixed models and generalized linear models. GLMMs can be developed for non-normally distributed responses, will allow nonlinear links between the mean of the response and the predictors, and can model over dispersion and associate by incorporating random effects (Breslow NE, Clayton DG. 1993).

In both generalized linear models (GLMs) and generalized linear mixed models (GLMMs), the emphasis is typically on estimating the mean structure, assuming that dispersion remains constant. However, real-world data can sometimes show more variability than what is predicted by the standard mean-variance relationship. When dispersion actually varies, relying on a constant dispersion assumption can significantly reduce the efficiency of mean parameter estimates [36]. To address this, a more flexible approach involves jointly modeling both the mean and the dispersion, allowing dispersion to be expressed as a function of covariates [37].

This study addressed the facts related with the trends, factors and model performances by focusing on the response variable, which is the maternal age at first birth in Ethiopia. Given the nature of the data, Gamma distribution was used for analysis. By leveraging the GLM, GLMM and distributional Gamma regression modelling framework with a log link function, it became feasible to establish a linear relationship between the predictors and the response variable. The log link function is used in Gamma regression because it ensures that predicted values remain strictly positive, aligning with the domain of the Gamma distribution. It transforms the nonlinear relationship between predictors and the response variable into a linear one on the log scale, facilitating interpretation and estimation. This transformation allows model coefficients to be interpreted multiplicatively, meaning a unit change in a predictor corresponds to a percentage change in the response. Additionally, the log link accommodates the Gamma distribution's variance structure, where variance increases with the square of the mean, and it enhances numerical stability during model fitting by preventing negative predictions and improving convergence. This approach, allows for analysis and interpretation of the factors influencing maternal age at first birth in the Ethiopian context.

**Gamma regression**

The gamma distribution is commonly used to represent continuous data that is restricted to positive values. When a variable ($y$) follows a gamma distribution ($y$, $\alpha$, $\beta$), its probability density function is determined by its mean $\mu$ ($\mu > 0$), shape parameter $\alpha$ ($\alpha > 0$), and rate parameters $\beta$ ($\beta > 0$). The probability density function for the gamma distribution is outlined by [37] as follows:

$$f(y) = \frac{\beta^{\alpha}}{\Gamma(\alpha)}y^{\alpha-1}e^{-\beta y}$$

(1)

Where:

$y$: Dependent variable.

$\Gamma(\alpha)$: Gamma function is defined as $\int_0^\infty y^{\alpha-1}e^{-y}dy$

Let $\alpha = v$ and $\beta = v/\mu \Rightarrow \mu = v/\beta$

$$f(y) = \frac{y^{-1}}{\Gamma(v)}\left(\frac{yv}{\mu}\right)^v e^{-\frac{yv}{\mu}}; y > 0 \tag{2}$$

Then $y \sim G\left(v, \frac{v}{\mu}\right)$. In this case $E(y) = \frac{\alpha}{\beta} = \frac{v}{\frac{v}{\mu}} = \mu$ and $var(y) = \frac{\alpha}{\beta^2} = \frac{v}{\frac{v}{\mu}^2} = \frac{\mu^2}{v}$

If $(y)$ has density function (2), we write $y \sim (\mu, v)$ [37]. Then the gamma regression model is defined as follows:

$$\mu_i = e^{(x_i'\beta_i + \epsilon)} \tag{3}$$

where $\epsilon$ is vector of random error with a dimension (n × 1), and (n) is the sample size.

One way to transform the gamma regression function to a linear form is to use the logarithm of the mean ($\mu_i$) as the response variable. This can be achieved by taking the natural logarithm of both sides of the equation (3) [38].

In order to analyze the determinants influencing the maternal age at first birth, three models were considered in GLM framework: The generalized linear model (GLM) (model 1) presented in equation (4), generalized linear mixed model (GLMM) (model 2), being in the presence of random effects and finally distributional Gamma regression model (model 3).

The GLM (model 1) generalizes a linear model and is composed of two fundamental elements. The first is the link function between the expected value of the maternal age at first birth and the linear predictors, and the second one is the variance function, in which the variance is expressed as a function of the mean. To explain the structure of the GLM, the linear model starts from:

$$\hat{y}_i = \eta_i = g(\mu_i) = \ln(\mu_i) = \mathbf{X}_i'\hat{\beta} \tag{4}$$

where $x_i$ is the $i^{th}$ row of $\mathbf{X} = [x_{i1}, x_{i2}, \ldots, x_{ip}]'$ of $\mathbf{X}$ which is an (n × p) data matrix with (p) explanatory variables, $\boldsymbol{\beta}$ is a (p × 1) vector of the slope coefficients and $\eta_i = g(\mu_i)$ is the log link function for the gamma regression model [38].

The second model applied was the GLMM (model 2) presented in equation (5). The GLMM is an extension of the GLM. The GLM includes in the mathematical model only the fixed effects, estimating their influence on the dependent variable. Instead, the GLMM considers, in addition to the fixed effects, the random effects. The random effects of the GLMM were the individual mothers in different enumeration areas, since the variability of maternal age was depend on the intrinsic characteristics of the mothers. In the gamma regression model using GLMM framework presented by [38], the shape parameter is constant for every observation, and it is modeled like the mean.

Here, we assume an independent random variables, $y_i \sim$ Gamma($\mu_i$; $v/\mu_i$); $i = 1, \ldots\ldots, n$, with mean parameter given by

$$\eta_i = g(\mu_i) = \mathbf{X}_i'\beta + \mathbf{Z}_i'\boldsymbol{b} \tag{5}$$

where $\beta = (\beta_1, \ldots\ldots\ldots\beta_p)'$ is fixed effect coefficients and $b = (b_1, \ldots\ldots\ldots\ldots b_k)'$, are the sets of random effect coefficients, $X_i$ and $Z_i$ are observed values of the fixed and random covariates respectively, and $\eta_i$ is the linear predictors at the $i^{th}$ observation. Again, it is convenient to take the first component of the $\boldsymbol{X}_i$ vector as 1 to allow for an intercept in the mean regression structure.

Sometimes modeling each parameters of the distribution parameters with covariates can be more informative than the GLM and GLMM application [39]. Using distributional Gamma regression instead of standard GLM (Generalized Linear

Model) or even GLMM (Generalized Linear Mixed Model) is beneficial when both the mean and dispersion (variability) of the outcome are systematically influenced by covariates.

In the classical parametric regression, both the mean and shape (inverse of dispersion) are modeled as functions of covariates via log-link functions as expressed in equation (6) and (7).

$$\text{Mean sub-model (location)}: \eta_i = g(\mu_i) = \mathbf{X}'_i\boldsymbol{\beta} \tag{6}$$

$$\text{Shape sub-model (inverse dispersion)}: g(\phi_i) = \mathbf{Z}'_i\gamma \tag{7}$$

$X_i$ is a vector of covariates (e.g., region, religion, residence) for observation $\beta$ is the vector of regression coefficients for the mean, $Z_i$ is a vector of covariates (may be same or different from $X_i$) for the shape, and $\gamma$ is the vector of regression coefficients for the shape. Parameter estimates $\beta$ and $\gamma$ are obtained via Maximum Likelihood Estimation (MLE) and Bayesian method.

## Maximum likelihood estimation

Adekanmbi V. (2017) [38] propose a classic approach to fit gamma regression model, where mean parameter follow regression structures in constant shape parameter, using the Fisher scoring algorithm. In that work, it was shown that for gamma re-parameterization given by (2) the GLM likelihood function can be written in the form:

$$L(\beta, \mathbf{y}) = \prod_{i=1}^{n} \frac{y_i^{v-1}}{\Gamma(v)} \left(\frac{v}{\mu_i}\right)^v e^{-\frac{vy_i}{\mu_i}} \tag{8}$$

and the log-likelihood function by:

$$L(\beta) = \sum_{i=1}^{n} \left\{ v\log(v) + (v-1)\log(y_i) - \log(\Gamma(v)) - v\log(\mu_i) - \frac{v}{\mu_i}y_i \right\} \tag{9}$$

Thus, assuming the regression structures defined by $\mu_i = e^{\mathbf{X}'_i\boldsymbol{\beta}}$ the score statistics are given by:

$$\frac{\partial L(\beta)}{\partial \beta_j} = v\sum_{i=1}^{n} x_{ij}\left(1 - \frac{y_i}{\mu_i}\right) ; j = 1,\ldots\ldots.p \tag{10}$$

and the Hessian matrix is determined by:

$$\frac{\partial^2 L(\beta)}{\partial \beta_k \partial \beta_j} = -v\sum_{i=1}^{n} x_{ij}x_{ik} * \frac{y_i}{\mu^2_i} ; j, k = 1,\ldots\ldots.p \tag{11}$$

In matrix form $H(\beta) = v\sum_{i=1}^{n} \frac{y_i}{\mu^2_i} x_{ij}x^T_{ik}$

Thus, the Fisher information matrix is given by:

$$-E\left(\frac{\partial^2 L(\beta)}{\partial \beta_k \partial \beta_j}\right) = v\sum_{i=1}^{n} \frac{y_i}{\mu^2_i} x_{ij}x_{ik} ; j, k = 1,\ldots\ldots.p \tag{12}$$

Similarly for the case of Model 2: Gamma GLMM (Fixed plus Random Effects), when the liner predictor become $\eta_i = g(\mu_i) = X'_i\beta + Z'_ib$ and random effect $b \sim N(0, D)$ the conditional likelihood become:

$$L(\beta, b, y) = \prod_{i=1}^{n} \frac{v^v}{\Gamma(v)} \left(y^{v-1}_i\right) \left(\frac{v}{\mu_i}\right)^v e^{-\frac{vy_i}{\mu_i}} * e^{-\frac{1}{2}b^T D^{-1} b}$$

(13)

Log-likelihood function become

$$L(\beta, b) = \sum_{i=1}^{n} \left[ v\log(v) - \log(\Gamma(v)) + (v-1)\log(y_i) - v\log(\mu_i) - \frac{v}{\mu_i}y_i \right] - \frac{1}{2}b^T D^{-1} b$$

(14)

The Hessian matrix become

$$H = \begin{bmatrix} H_{\beta\beta} & H_{\beta b} \\ H_{b\beta} & H_{bb} \end{bmatrix}$$

(15)

Where

$$H_{\beta\beta} = -v\sum_{i=1}^{n} \frac{y_i}{\mu^2_i} x_i x^T_i,$$

$$H_{\beta b} = -v\sum_{i=1}^{n} \frac{y_i}{\mu^2_i} x_i z^T_i \text{ and}$$

$$H_{bb} = -v\sum_{i=1}^{n} \frac{y_i}{\mu^2_i} z_i z^T_i - D^{-1}$$

It can be noted that the Fisher information matrix is a block diagonal matrix, where one of the blocks corresponds to the mean regression parameters $\beta$ and the other to the random effect $b$. Thus, $\beta$ and $b$ are orthogonal and their maximum likelihood estimators $\hat{\beta}$ and $\hat{b}$ are asymptotically independent. As a consequence of this result, (Adekanmbi V., 2017) [38] proposed an iterative algorithm to obtain the maximum likelihood estimates of the regression parameters.

In the similar way the distributional gamma regression model (model 3) the likelihood function is derived as follows

$$f(y_i \mid \mu_i, \phi_i) = \frac{\phi_i^{\phi_i}}{\Gamma(\phi_i)\mu_i^{\phi_i}} y_i^{\phi_i-1} \exp\left(-\frac{y_i\phi_i}{\mu_i}\right)$$

(16)

Where $\mu_i$ and $\phi_i$ are the location and shape (inverse of dispersion) respectively and those expressed as regression coefficient as follows

$$\mu_i = e^{X'_i\beta}$$

$$\phi_i = e^{Z'_i\gamma}$$

The log-likelihood for a sample of n independent observations is

$$\ell(\mu, \phi) = \sum_{i=1}^{n} \left[ \phi_i log(\phi_i) - log\Gamma(\phi_i) + (\phi_i-1)logy_i - \phi_i log(\mu_{i\circ}) - \frac{y_i\phi_i}{\mu_{i\circ}} \right] \tag{17}$$

Finally we can derive the score function (first derivatives) and the Hessian matrix (second derivatives) and fisher information for parameters $\mu_i$ and $\phi_i$ we assume $\mu_i > 0$, $\phi_i > 0$, and $y_i > 0$.

$$\ell(\beta, \gamma) = \sum_{i=1}^{n} \left[ \mathbf{Z}'_i\gamma e^{\mathbf{Z}'_i\gamma} - log\Gamma\left(e^{\mathbf{Z}'_i\gamma}\right) + \left(e^{\mathbf{Z}'_i\gamma} - 1\right) log\mathbf{y}_i - \mathbf{X}'_i\beta e^{\mathbf{Z}'_i\gamma} - \frac{\mathbf{y}_i e^{\mathbf{Z}'_i\gamma}}{e^{\mathbf{X}'_i\beta}} \right]$$

$$\frac{\partial\ell(\beta, \gamma)}{\partial\beta} = \sum_{i=1}^{n} \frac{\phi_i(\mathbf{y}_i-\mu_i)}{\mu_i} X_i$$

$$\frac{\partial\ell(\beta, \gamma)}{\partial\gamma} = \sum_{i=1}^{n} \left[ log(\phi_i) + 1 - \Psi(\phi_i) + log\mathbf{y}_i - log(\mu_i) - \frac{y_i}{\mu_i} \right] \phi_i Z_i$$

Where $\Psi(\phi_i)$ is digamma function (the first derivatives of $log\Gamma(\phi_i)$)

The Hessian matrix become

$$H_{\beta\beta} = \sum_{i=1}^{n} \left[ \frac{\mathbf{y}_i\phi_i}{\mu_i} \right] \mathbf{X}_i\mathbf{X}'_i$$

$$H_{\gamma\gamma} = \sum_{i=1}^{n} \phi^2_i\Psi^1(\phi_i) \mathbf{Z}_i\mathbf{Z}'_i$$

$\Psi^1(\phi_i)$ is trigamma function (the first derivatives of digamma function)

$$H_{\gamma\beta} = H_{\beta\gamma} = \sum_{i=1}^{n} \phi_i \left[ \frac{\mathbf{y}_i-\mu_i}{\mu_i} \right] \mathbf{X}_i\mathbf{Z}'_i$$

After developing such mathematical derivation we can estimate the values of the parameters vector $\beta$ and $\gamma$ using iterative optimization in Newton-Raphson algorithm.

## Bayesian estimation

Combining the prior probability density function of the parameters with the likelihood function of the observations yields the Bayesian estimator, which is only the information-rich part of the posterior probability density function [40]. The Bayesian method to fit gamma regression models, where mean parameter follow regression structure. As in these works, to implement a Bayesian approach to estimate the parameters of the gamma regression model, we need to specify a prior distribution for the parameters and the data (Y) likelihood and relate to the posterior density. The relation become:

$$\Pi(\beta/data) = \frac{L(data/\beta)P(\beta)}{\int L(data/\beta)P(\beta)} \tag{18}$$

Where, $\mathbf{\Pi}\left(\beta|data\right)$ is the posterior distribution of the parameters $\beta$ and given the data, $P\left(Y|\theta\right)$ is the likelihood of the data given the parameters, $P(\beta)$ is the prior distribution of the parameters.

The first procedure to formulate the posterior distribution is specifying the likelihood function in both GLM and GLMM framework. For the case of GLM the likelihood function is formulated in Equation (6) as

$$L\left(\beta, \boldsymbol{y}\right) = \prod_{i=1}^{n} \frac{y_i^{v-1}}{\Gamma(v)} \left(\frac{v}{\exp(x'_i\beta)}\right)^v \exp\left(-\frac{vy_i}{\exp(x'_i\beta)}\right) \tag{19}$$

and for GLMM the likelihood function is formulated in equation (11) as

$$L\left(\beta, \boldsymbol{b}, \boldsymbol{y}\right) = \prod_{i=1}^{n} \frac{v^v}{\Gamma(v)} \left(y^{v-1}_i\right) \left(\frac{v}{\exp\left(X'_i\beta + Z'_ib\right)}\right)^v \exp\left(-\frac{vy_i}{\exp\left(X'_i\beta + Z'_ib\right)}\right) * e^{-\frac{1}{2}\boldsymbol{b}^T D^{-1}b} \tag{20}$$

Secondly, fix the prior distribution depending on the data nature or by highlighting previous literature about maternal age at first birth specifically in Ethiopia. Mathematically the prior for GLM Gamma regression model coefficients a common choice is a multivariate normal prior for the regression coefficients:

$$\beta \sim \boldsymbol{N}(\mu_\beta, \Sigma_\beta)$$

$$\boldsymbol{p}\left(\beta\right) = \frac{1}{\left(2\pi\right)^{p/2} |\Sigma_\beta|^{1/2}} \boldsymbol{exp}\left(-\frac{1}{2}\left(\beta-\mu_\beta\right)^T \Sigma_\beta^{-1}\left(\beta-\mu_\beta\right)\right) \tag{21}$$

And for GLMM case the fixed prior and the random effect prior is needed

$$\boldsymbol{b} \sim \boldsymbol{N}(\boldsymbol{0}, \boldsymbol{D})$$

$$\boldsymbol{p}\left(\beta\right) = \frac{1}{\left(2\pi\right)^{q/2} |D|^{1/2}} \boldsymbol{exp}\left(-\frac{1}{2}\boldsymbol{b}^T D^{-1}\boldsymbol{b}\right) \tag{22}$$

For the intercept $\beta_0 \sim N(3, 0.5)$ based on the data nature and previous literature, and the other fixed coefficients $\boldsymbol{\beta} \sim N(0, 1)$.

Lastly using equations (15–18) the expressions for the likelihood and priors [41], the full posterior distribution for Gamma GLM regression can be formulated as

$$\mathbf{\Pi}\left(\beta/data\right) = \prod_{i=1}^{n} \frac{y_i^{v-1}}{\Gamma(v)} \left(\frac{v}{\exp(x'_i\beta)}\right)^v \exp\left(-\frac{vy_i}{\exp(x'_i\beta)}\right) * \frac{1}{(2\pi)^{p/2}|\Sigma_\beta|^{1/2}} \exp\left(-\frac{1}{2}\left(\boldsymbol{\beta}-\mu_\beta\right)^T \Sigma_\beta^{-1}\left(\boldsymbol{\beta}-\mu_\beta\right)\right) \tag{23}$$

The extended posterior by the effect of random component become

$$\left(\beta/data\right) = \prod_{i=1}^{n} \frac{y_i^{v-1}}{\Gamma(v)} \left(\frac{v}{\exp(x'_i\beta)}\right)^v \exp\left(-\frac{vy_i}{\exp(x'_i\beta)}\right) * \frac{1}{(2\pi)^{\frac{p}{2}} |\Sigma_\beta|^{\frac{1}{2}}} \exp\left(-\frac{1}{2}\left(\boldsymbol{\beta}-\mu_\beta\right)^T \Sigma_\beta^{-1}\left(\boldsymbol{\beta}-\mu_\beta\right)\right)$$
$$* \frac{1}{(2\pi)^{q/2}|D|^{1/2}} \exp\left(-\frac{1}{2}\boldsymbol{b}^T D^{-1}\boldsymbol{b}\right) \tag{24}$$

The mathematical formulation of the Bayesian estimation for the joint modeling of mean and dispersion in gamma regression, the likelihood function is:

$$f(y_i \mid \mu_i, \phi_i) = \frac{\phi_i^{\phi_i}}{\Gamma(\phi_i)\,\mu_i^{\phi_i}} y_i^{\phi_i - 1} \exp\left(-\frac{y_i \phi_i}{\mu_i}\right)$$

(25)

By assuming the normal prior for the regression coefficients

$$\beta \sim N(b_0, B_0)$$

(26)

$$\gamma \sim N(g_0, G_0)$$

(27)

Using Bayes' theorem, the joint posterior is expressed using equation (25), (26) and (27)

$$\prod_{i=1}^{n}\left[\frac{\phi_i^{\phi_i}}{\Gamma(\phi_i)\,\mu_i^{\phi_i}} y_i^{\phi_i - 1} \exp\left(-\frac{y_i \phi_i}{\mu_i}\right) \exp\left(-\frac{1}{2}(\beta - b_0)^T B_0^{-1}(\beta - b_0)\right) \exp\left(-\frac{1}{2}(\gamma - g_0)^T G_0^{-1}(\gamma - g_0)\right)\right]$$

(28)

**Stan Modeling:** Stan is a specialized state dialect implemented as a C++ library designed for Bayesian modeling, as noted by [41]. It serves as a Bayesian programming tool that primarily employs the No-U-Turn Sampler (NUTS) to perform posterior simulations based on user-defined models and data. Hamiltonian Monte Carlo (HMC), is a method that falls under the broader category of Markov Chain Monte Carlo (MCMC) techniques. While HMC can be quite intricate due to the need for calculating potentially complex derivatives and fine-tuning several parameters, it often proves to be faster and more reliable than simpler methods like basic Markov chain simulation, Gibbs sampling, and the Metropolis algorithm, particularly in more complex scenarios, as it navigates the posterior parameter space more effectively.

They achieve this by associating each model parameter with a momentum variable, which influences the exploration behavior of Hamiltonian Monte Carlo (HMC) based on the posterior density of the currently sampled parameter. This approach allows HMC to reduce the random walk characteristics typical of the Metropolis algorithm [41]. As a result, Stan demonstrates significantly greater efficiency compared to conventional Bayesian software algorithms like Metropolis and Gibbs. Notably, the primary function of the rstan package is to interface with the Stan software to estimate a given statistical model, and rstan features an intelligent system that automates much of the adaptation process.

A statistical model can be represented through a conditional probability function P(β, δ | y, x), where (β, δ) denotes a series of unknown values being modeled, y represents a series of known response variable values, and x includes un modeled predictors and constants, such as sizes and hyper parameters. A Stan program explicitly defines a log probability function for parameters based on the provided data and constants. Stan facilitates comprehensive Bayesian inference for continuous-variable models using Markov chain Monte Carlo methods, including an enhanced version of Hamiltonian Monte Carlo sampling [42]. Users can access Stan from R via the rstan package. This interfaces support sampling and optimization-based inference, along with diagnostics and posterior analysis. Additionally, rstan offer functionalities for accessing log probabilities, transforming parameters, and creating specialized plots. Stan programs are structured with variable type declarations and statements, which include constrained and unconstrained integers, scalars, vectors, and matrices. Variables are organized into blocks that correspond to their intended use: data, transformed data, parameters, transformed parameters, or generated quantities.

**Model Diagnosis Checking:** model 1, model 2 and distributional gamma regression were specified as in the gamma family and with log link. For the case of maximum likelihood estimation method (MLE), the goodness of fit for GLM, GLMM

and distributional Gamma regression was checked by the Akaike information criterion (AIC), Bayesian information criterion (BIC) and log likelihood test; the best model is the one that minimizes these values. For the case of Bayesian technique the goodness of fit the performance of each model was evaluated by using potential scale reduction factor ($\hat{R}$) statistic, effective sample size value, Monte Carlo standard error (MCSE) and posterior predictive checking method. Loo method with expected log pointwise predictive density and their standard errors were used for comparison of GLM, GLMM and distributional Gamma regression.

## Results

### Descriptive characteristics of the study sample

A total of 5,839 women aged 15–49 years who had experienced at least one live birth were included in this analysis using data from the 2019 Ethiopia Mini Demographic and Health Survey. Among these women, the mean maternal age at first birth was 18.32 years. Maternal age at first birth among the respondents ranged from 10 years to 40 years, indicating substantial variability in the timing of entry into motherhood among women who had already experienced first birth at the time of the survey.

Table 1 presents the geographic, residential, and religious characteristics of the study participants. The regional distribution of respondents ranged from 6.4% in Addis Ababa to 12.1% in Oromia. The largest proportions of participants were from Oromia (12.1%), SNNPR (11.8%), and Amhara (11.0%), reflecting the relatively larger populations of these regions in Ethiopia. With respect to place of residence, 71.8% of respondents lived in rural areas, while 28.2% resided in urban areas. This distribution reflects the predominantly rural composition of Ethiopia's population. Regarding religious affiliation, Muslim women represented the largest group (42.8%), followed by Orthodox Christians (35.3%) and Protestants (19.9%).

Smaller proportions were observed for Catholics (0.6%), traditional religion followers (1.1%), and other religions (0.3%). These descriptive statistics provide an overview of the socio-demographic composition of the analytical sample consisting of women who had already experienced first birth at the time of the survey.

Fig 1 presents the histogram and density plot of maternal age at first birth among women who had experienced first birth. The distribution is positively skewed, with a concentration of observations in the late teenage years and a longer right tail extending toward older ages. As expected, the outcome variable takes only positive values, consistent with the nature of age-at-event data.

Fig 2 illustrates boxplots of maternal age at first birth across selected covariate categories, including region and religion. The boxplots suggest variability in the distribution of maternal age at first birth across these groups. A small number of outliers are observed in several categories, reflecting cases of unusually early or late first births.

### Comparison of candidate distributions

Fig 3 compares the fitted Gamma, Normal, log-normal and Weibull distributions to the observed maternal age at first birth of women who had already give birth in Ethiopia. The gamma distribution (blue curve) provides the best visual fit to the observed histogram, capturing both the central peak around the late teens and the gradual decline toward older ages. Its flexibility in modeling positively skewed, non-negative data makes it well-suited for maternal age at first birth, where the variable cannot take values below puberty and tends to have a longer right tail. The gamma fit closely follows the observed frequency in both the lower and middle age ranges, and its right-tail decay is consistent with the rarity of very late first births. This close alignment suggests that the gamma distribution's shape and scale parameters effectively represent the underlying variability in the data.

The normal distribution (red curve) captures the general center of the distribution but performs less well in the tails. Because the normal is symmetric, it slightly overestimates the frequency of very low and very high ages and underestimates the skewness present in the data. In contrast, the log-normal distribution (green curve) show discrepancies which implies the log-normal less appropriate for this dataset compared to the gamma, despite its theoretical appeal for positively skewed variables.

**Table 1. Geographic, residential and religious characteristics of study participants, extracted from Mini EDHS 2019 survey dataset (Ethiopia), that conducted from March 21, 2019, to June 28, 2019 (N = 5839).**

| Variables | Categories | frequency | % |
|---|---|---|---|
| **region** | Tigray | 501 | 8.6 |
| | Afar | 500 | 8.6 |
| | Amhara | 643 | 11.0 |
| | Oromia | 707 | 12.1 |
| | Somali | 428 | 7.3 |
| | Benishangul-Gumuz | 523 | 9.0 |
| | SNNPR | 692 | 11.8 |
| | Gambela | 529 | 9.1 |
| | Harari | 476 | 8.1 |
| | Addis Ababa | 374 | 6.4 |
| | Dire Dewa | 469 | 8.0 |
| **Residence** | Urban | 1645 | 28.2 |
| | Rural | 4197 | 71.8 |
| **Religion** | Orthodox | 2060 | 35.3 |
| | Catholic | 20 | 0.6 |
| | Protestant | 1165 | 19.9 |
| | Muslim | 2498 | 42.8 |
| | Traditional | 45 | 1.1 |
| | Others | 17 | 0.3 |

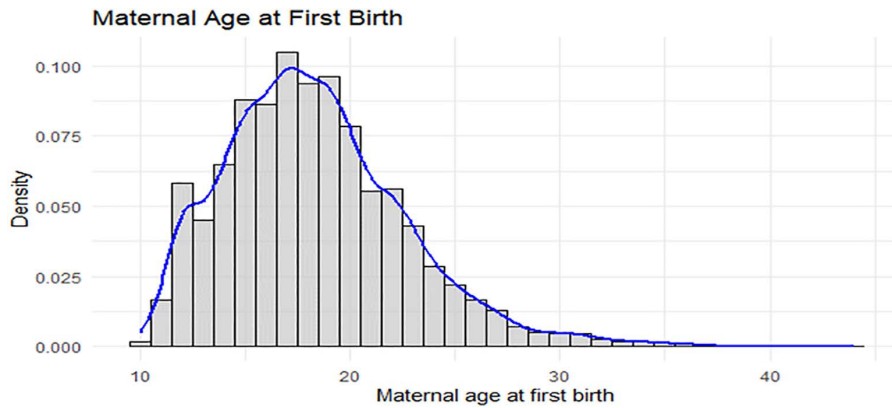

**Fig 1. Maternal Age at first birth histogram and density.**

The Weibull distribution (purple curve) offers a moderately good fit, particularly in the lower to mid-range of ages, but it underestimates the peak and slightly overshoots in the left tail. While the Weibull is also flexible and can accommodate skewed, non-negative data, its parameterization here does not align as closely with the observed pattern as the gamma does. This may be because the Weibull's hazard-based shape is less suited to the gradual tail decay of maternal age at first birth, compared to the gamma's smoother decline. The gamma emerges as the most balanced fit across the entire range of ages in this dataset.

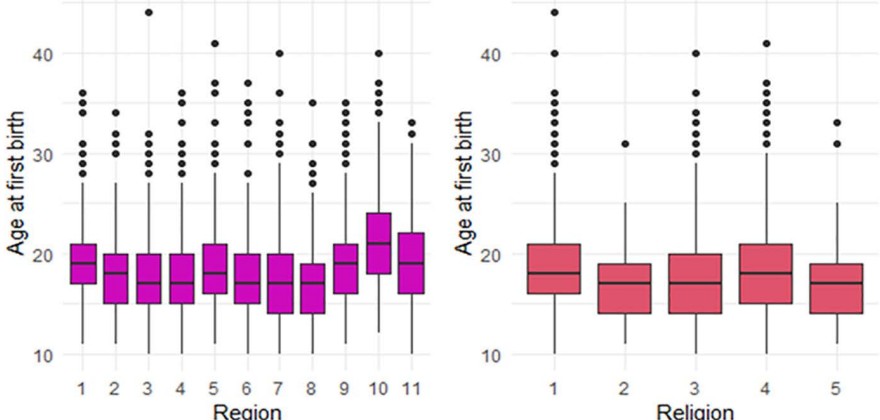

**Fig 2. Maternal age at first birth box plot on regions and religion.**

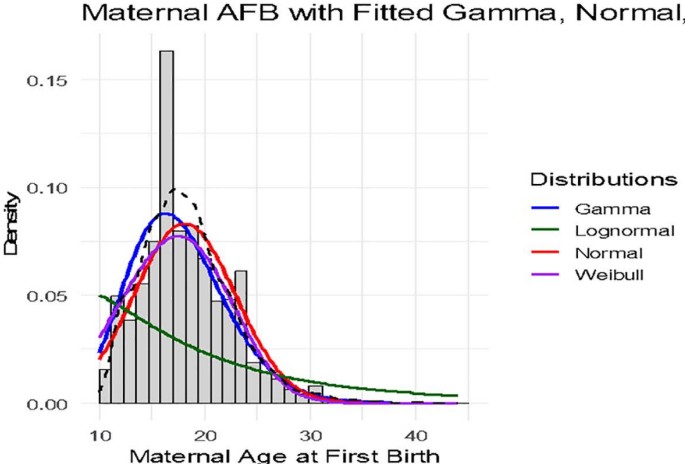

**Fig 3. Maternal age at first birth curve (black) with gamma and normal distribution curves.**

Given these comparisons, the gamma distribution emerges as the most appropriate model for maternal age at first birth in this dataset, offering a close empirical fit while aligning with the theoretical expectation that such demographic events follow a positively skewed, non-negative distribution. Its flexibility in capturing variation across regions, religions, and residential contexts supports our conceptual framework, which emphasizes the interplay of structural and cultural factors in shaping the timing of first births. By adopting the gamma distribution, our subsequent regression analyses can more accurately reflect the underlying data structure, thereby providing more reliable estimates of the associations between key socio-demographic predictors and maternal age at first birth.

The result in Table 2 supports the evidence in Fig 4 that Gamma distribution emerged as the strongest candidate for modeling the maternal age at first birth data, as indicated by its lowest AIC (−5427.93) and BIC (−5051.38) values among the four candidate models. Its log-likelihood (−2776.97) was also the highest (least negative), suggesting superior fit to the data. The Kolmogorov–Smirnov (K–S) statistic for the Gamma model was 0.14 with a p-value of 0.292, which is above

**Table 2. Model Comparison for Maternal Age at First Birth Data Using Information Criteria and Goodness-of-Fit Statistics.**

| Models | AIC | BIC | Likelihood | K-S | P-value |
|---|---|---|---|---|---|
| **Gamma** | **−5427.932** | **−5051.38** | **−2776.97** | **0.14** | **0.292** |
| Normal | −5341.218 | −4951.23 | −2720.89 | 0.112 | 0.054 |
| Weibull | −5303.092 | −5099.876 | −2685.55 | 0.097 | 0.050 |
| Log_normal | −5235.748 | −4981.00 | −2737.79 | 0.115 | 0.041 |

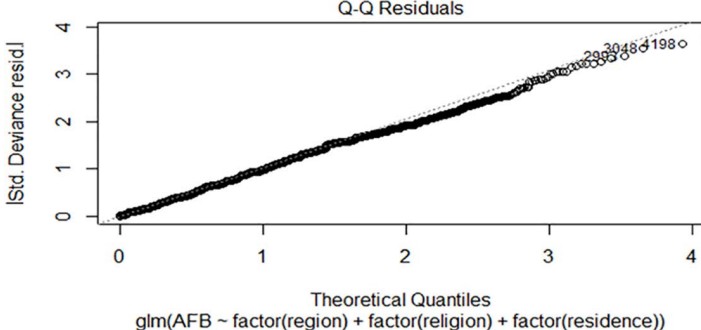

**Fig 4. Residual diagnosis.**

the conventional significance threshold of 0.05, indicating no evidence to reject the null hypothesis that the data follow this distribution. This combination of excellent information criteria performance and acceptable goodness-of-fit test results provides strong empirical support for the Gamma model's suitability.

The Normal distribution performed less favorably, with higher AIC (−5341.22) and BIC (−4951.23) values compared to the Gamma model. While its K–S statistic (0.112) was slightly lower than Gamma's, the associated p-value (0.034) fell below 0.05, suggesting a statistically significant departure from normality in the data. This aligns with theoretical expectations, as maternal age at first birth data are often positively skewed, a feature that the symmetric Normal distribution cannot adequately capture. Consequently, despite being a common baseline model, the Normal distribution appears unsuitable for this application. The Log-normal and Weibull distributions showed intermediate performance between Gamma and Normal. The Weibull model achieved lower AIC (−5303.09) and BIC (−5099.88) than Log-normal, and both had K–S p-values (0.041 for Log-normal, 0.050 for Weibull) near or at the significance threshold, suggesting marginal fits. While Weibull model can accommodate skewness better than the Normal distribution, it still failed to match the fit quality of the Gamma model, as reflected in their higher AIC/BIC values and slightly less favorable K–S results.

## Diagnosis checking

In the maximum likelihood estimation within gamma regression, model diagnosis hinges on scrutinizing gamma regression residuals. This analytical process is geared towards pinpointing outliers and potential model misspecifications.

Assessing the distribution of residuals through a Quantile-Quantile (QQ) plot, as described in Fig 4, reveals that the residuals closely mimic a normal distribution. This alignment suggests that the residuals follow a normal distribution pattern, a favorable situation in model evaluation.

## Model performance and model comparison

To evaluate the adequacy of alternative modeling frameworks for maternal age at first birth (AFB), three Gamma-based regression models were estimated: the Generalized Linear Model (GLM), the Generalized Linear Mixed Model (GLMM), and a Distributional Gamma regression model that allows covariates to influence both the mean and dispersion parameters of the outcome distribution. All models assumed that maternal age at first birth follows a Gamma distribution, which is appropriate for modeling positive, continuous, and right-skewed variables.

Each model was estimated using both classical maximum likelihood estimation (MLE) and Bayesian inference, enabling comparison of model fit and predictive performance across estimation frameworks. Model comparison criteria for the classical models included the Akaike Information Criterion (AIC), Bayesian Information Criterion (BIC), and the log-likelihood, while the Bayesian models were evaluated using the Widely Applicable Information Criterion (WAIC), leave-one-out cross-validation (LOO), and the Deviance Information Criterion (DIC).

### Classical model comparison

Under the classical estimation framework, the distributional Gamma regression model demonstrated the best overall fit among the competing models (Table 3). This model achieved the lowest AIC (−72,375.6) and lowest BIC (−71,245.3) values, indicating superior performance when balancing model fit and complexity. The model also produced the highest log-likelihood value, suggesting improved agreement with the observed data relative to the standard GLM and GLMM specifications.

The improved performance of the distributional Gamma model likely reflects its ability to model both the conditional mean and the dispersion of maternal age at first birth simultaneously, thereby capturing heterogeneity in the variability of the outcome across different socio-demographic groups.

### Bayesian model comparison

Under the Bayesian estimation framework, model comparison based on WAIC, LOO cross-validation, and DIC produced consistent results. The Bayesian distributional Gamma regression model achieved the lowest values for all three criteria (WAIC = −1870, LOO = −1876, and DIC = −1865), indicating the best predictive performance among the candidate models. These results suggest that allowing covariates to influence both the location (mean) and dispersion components of the Gamma distribution improves the model's ability to capture variability in maternal age at first birth among women who had already experienced first birth.

Taken together, the results from both classical and Bayesian comparisons indicate that the distributional Gamma regression model provides the best overall fit to the data. This model was therefore selected as the primary framework for interpreting the associations between maternal age at first birth and the explanatory variables considered in this study.

Although the distributional Gamma regression model demonstrated the strongest overall performance, estimating multiple model frameworks (GLM, GLMM, and distributional regression) provides additional insight into the robustness

**Table 3. Comparative Performance of Gamma-Based Models for Maternal Age at First Birth under Classical and Bayesian Frameworks.**

| Criteria | AIC | BIC | Log_lik | WAIC | LOO | DIC |
|---|---|---|---|---|---|---|
| GLM (MLE) | −32500 | −32606.71 | −16233.9 | --- | --- | --- |
| GLMM (MLE) | −70015.7 | −69902.3 | −35024.8 | --- | --- | --- |
| D. Gamma GLM (MLE) | −72375.6 | −71245.3 | −43426.5 | --- | --- | --- |
| GLM(Bayesian) | --- | --- | --- | −1860.3 | −1858.5 | −1850.4 |
| GLMM(Bayesian) | --- | --- | --- | −1862.1 | −1865.7 | −1860.2 |
| D.Gamma GLM (Bayesian) | --- | --- | --- | −1870 | −1876 | −1865 |

of the results. Comparing these models allows assessment of how different modeling assumptions such as accounting for unobserved heterogeneity or modeling dispersion affect the estimated associations between predictors and maternal age at first birth. This comparative approach helps evaluate the consistency of coefficient estimates, the stability of effect directions, and the predictive adequacy of the models, thereby strengthening confidence in the empirical findings while acknowledging the limitations inherent in observational cross-sectional data.

## Classical Gamma regression model

We recognize that excluding women who had not yet given birth, especially younger and potentially more socioeconomically advantaged individuals, introduces a form of selection bias. For example, if highly educated urban women tend to postpone childbearing beyond the survey period, their systematic omission could lead to underestimation of the association between urban residence, higher education, and delayed first birth. This truncation of the outcome distribution may therefore bias parameter estimates toward earlier ages. Our results should thus be interpreted as describing the determinants of age at first birth among women who have already had a child, rather than the determinants of timing for all women of reproductive age.

Table 4 presents the estimation results of both the Generalized Linear Model (GLM) and Generalized Linear Mixed Model (GLMM) using a gamma distribution with a log link. Both models produced similar coefficient estimates, reflecting consistency in the direction and magnitude of effects.

After adjusting for region and religion, rural women who had already given birth had an expected age at first birth 4.5% lower than that of their urban counterparts ($p < 0.05$). Controlling for region and residence, mothers who had previously given birth and identified as Orthodox or Protestant exhibited significantly higher average maternal ages at first birth by 4% and 3%, respectively compared to Muslim mothers (the reference group), with results statistically significant at

**Table 4. Parameter Estimates of Maternal Age at First Birth from Gamma Regression Models (GLM and GLMM) with Geographic, Residential and Religious Covariates.**

| Variables | categories | GLM | GLMM | Sig |
|---|---|---|---|---|
| | | Estimates | Estimates | |
| Intercept | | 2.9724 | 2.912 | <2e-16 |
| Residence | Urban | Base | | |
| | Rural | −0.04589 | −0.04633 | 2.95e-09 |
| Religion | Muslim | | | |
| | Orthodox | 0.040187 | 0.040248 | 1.60e-06 |
| | Protestant | 0.033693 | 0.034783 | 1.85e-07 |
| | Others | −0.07781 | −0.07783 | 0.73823 |
| Region | Tigray | | | |
| | Afar | −0.0456 | −0.0521 | 0.977476 |
| | Amhara | −0.04666 | −0.04944 | 0.000355 |
| | Oromia | −0.03204 | −0.03214 | 0.024874 |
| | Somali | 0.039349 | 0.037485 | 0.018055 |
| | SNNPR | −0.02819 | −0.03426 | 0.052573 |
| | Benishangul | −0.03178 | −0.03760 | 0.032975 |
| | Gambela | −0.08425 | −0.08426 | 4.88e-08 |
| | Hareri | 0.016448 | 0.016669 | 0.296946 |
| | Addis Ababa | 0.105941 | 0.097785 | 1.60e-10 |
| | Dere-dawa | 0.038606 | 0.031785 | 0.014686 |

*p* < 0.05. In contrast, mothers in category others (Catholic, traditional, and other) religions had an expected maternal age at first birth that was 7.5% lower than that of Muslim mothers.

There are clear association in between different regions and maternal age at first birth (AFB) relative to Tigray. Women who had already had child in Amhara ($\beta = -0.047$, p < 0.001), Oromia ($\beta = -0.032$, p = 0.025), Benishangul ($\beta = -0.032$, p = 0.033), and Gambela ($\beta = -0.084$, p < 0.001) experienced significantly earlier first births, with Gambela showing the strongest negative effect. By contrast, women in Somali ($\beta = 0.038$, p = 0.018), Addis Ababa ($\beta = 0.10$, p < 0.001), and Dire Dawa ($\beta = 0.032$, p = 0.015) initiated childbearing at older ages, with Addis Ababa exhibiting the largest positive effect, suggesting substantially delayed AFB. Differences in Afar ($\beta = -0.046$, p = 0.977) and Harari ($\beta = 0.016$, p = 0.297) were not statistically significant, while SNNPR ($\beta = -0.028$, p = 0.053) showed borderline evidence of earlier first births compared to Tigray. These numerical patterns underscore the presence of strong geographic disparities in the timing of first birth in Ethiopia.

## Sensitivity analysis

To assess the robustness of the Gamma regression results and the potential impact of omitted socioeconomic variables, we re-estimated the model including maternal education, marital status, occupation, age at first marriage, and health service access alongside the original geographic, religious, and residential covariates.

As shown in Table 5, the inclusion of these additional predictors resulted in only marginal changes to the coefficients for region, religion, and residence, with all percentage changes in magnitude remaining below 10%. The statistical significance and direction of these effects were preserved. For example, the coefficient for rural residence changed from −0.04589 to −0.04112 (10.4%), while the coefficient for Addis Ababa decreased slightly from 0.10594 to 0.09987 (5.7%). Similarly, the coefficients for Orthodox and Protestant religious affiliation decreased by 3.5% and 5.0%, respectively, but remained statistically significant.

In the expanded model, several of the added socioeconomic covariates were themselves significant predictors of maternal age at first birth. Higher maternal education (secondary or above), later age at first marriage, marital status, occupation, and better health service access were all positively associated with older ages at first birth.

**Table 5. Sensitivity Analysis: Parameter Estimates of Maternal Age at First Birth from GLM Gamma Regression Models with and Without Additional Socioeconomic Variables.**

| Variable | Category | Baseline Model | Expanded Model | % Change in Estimate | Sig. (Expanded) |
|---|---|---|---|---|---|
| **Residence** | Rural | −0.04589 | −0.04112 | −10.4% | 4.21e-08 |
| **Religion** | Orthodox | 0.04019 | 0.03877 | −3.5% | 2.01e-06 |
| | Protestant | 0.03369 | 0.03201 | −5.0% | 3.15e-07 |
| | Others | −0.07781 | −0.07492 | −3.7% | 0.725 |
| **Region** | Afar | −0.04560 | −0.04306 | −5.6% | 0.977476 |
| | Amhara | −0.04666 | −0.04418 | −5.3% | 0.000498 |
| | Oromia | −0.03204 | −0.03022 | −5.7% | 0.026841 |
| | Somali | 0.03935 | 0.03713 | −5.6% | 0.019001 |
| | SNNPR | −0.02819 | −0.02652 | −5.9% | 0.053211 |
| | Benishangul | −0.03178 | −0.03001 | −5.6% | 0.033421 |
| | Gambela | −0.08425 | −0.07931 | −5.9% | 6.11e-08 |
| | Hareri | 0.01645 | 0.01549 | −5.8% | 0.297842 |
| | Addis Ababa | 0.10594 | 0.09987 | −5.7% | 2.48e-10 |
| | Dire Dawa | 0.03861 | 0.03639 | −5.7% | 0.015101 |

The best-performing specification identified in the model comparison (Table 3) was the distributional Gamma regression model, which simultaneously models the mean (location) and dispersion components of the outcome. The mean sub-model represents the expected maternal age at first birth, while the dispersion sub-model captures variability around that mean. This modeling framework is particularly useful because maternal age at first birth among women who had already experienced a first birth may vary not only in its expected value but also in the degree of variability across socio-demographic groups.

As shown in Table 6, the intercept in the mean sub-model is 2.929 ($p < 0.001$). Under the log link function, exponentiation this coefficient indicates that the reference group defined as urban Muslim women residing in Tigray has an expected maternal age at first birth of approximately 18.7 years. In the dispersion sub-model, the intercept is 3.534 ($p < 0.001$), representing the baseline level of variability in maternal age at first birth for the same reference group. These intercepts

**Table 6. Classical Gamma Regression Estimates of Mean and Dispersion Parameters for Maternal Age at First Birth by Region, Religion, and Residence in Ethiopia.**

| Variables | | Estimate | Std. Error | z value | Pr(>|z|) |
|---|---|---|---|---|---|
| (Intercept) | | 2.929146 | 0.013810 | 212.11 | <2e-16 |
| Region | Afar | 0.002030 | 0.015547 | 0.13 | 0.89614 |
| | Amhara | −0.044855 | 0.012213 | −3.67 | 0.00024 |
| | Oromia | −0.029181 | 0.013359 | −2.18 | 0.02893 |
| | Somali | 0.040656 | 0.015245 | 2.67 | 0.00766 |
| | SNNPR | −0.026723 | 0.013762 | −1.94 | 0.05217 |
| | Benishangul | −0.027383 | 0.014624 | −1.87 | 0.06114 |
| | Gambela | −0.080535 | 0.014303 | −5.63 | 1.79e-08 |
| | Hareri | 0.017556 | 0.015175 | 1.16 | 0.24731 |
| | Addis Ababa | 0.105571 | 0.015923 | 6.63 | 3.35e-11 |
| | Dere-dawa | 0.038628 | 0.014804 | 2.61 | 0.00907 |
| Religion | Orthodox | 0.043411 | 0.008842 | 4.91 | 9.12e-07 |
| | Protestant | 0.030317 | 0.010349 | 0.03 | 0.037556 |
| | Others | −0.048531 | 0.022435 | −0.38 | 0.03474 |
| Residence | Rural | −0.046430 | 0.007616 | −6.10 | 1.08e-09 |
| **Shape (inverse of dispersion)** | | | | | |
| (Intercept) | | 3.533875 | 0.094590 | 37.36 | <2e-16 |
| Region | Afar | −0.255548 | 0.103745 | −5.35 | 8.56e-08 |
| | Amhara | −0.184934 | 0.084218 | −5.05 | 4.52e-07 |
| | Oromia | −0.204783 | 0.091518 | −4.53 | 5.84e-06 |
| | Somali | −0.310920 | 0.106652 | −2.92 | 0.00355 |
| | SNNPR | −0.243026 | 0.093775 | −4.72 | 2.31e-06 |
| | Benishangul | −0.321268 | 0.095746 | −6.07 | 1.27e-09 |
| | Gambela | −0.276238 | 0.098224 | −2.81 | 0.00492 |
| | Hareri | −0.229073 | 0.100505 | −5.16 | 2.41e-07 |
| | Addis Ababa | −0.246986 | 0.104692 | −4.27 | 1.96e-05 |
| | Dere-dawa | −0.141977 | 0.100251 | −4.01 | 6.08e-05 |
| Religion | Orthodox | −0.149941 | 0.056833 | −2.64 | 0.00833 |
| | Protestant | −0.081633 | 0.066370 | −1.23 | 0.21871 |
| | Others | −0.152352 | 0.137822 | −1.11 | 0.26898 |
| Residence | Rural | −0.067614 | 0.049911 | −0.15 | 0.03875 |

provide the baseline against which the effects of the explanatory variables on both the expected age and its variability can be evaluated.

After adjusting for religion and place of residence, notable regional differences in maternal age at first birth remain evident. Relative to Tigray, mothers who had already experienced a first birth in Addis Ababa, Somali, and Dire Dawa had significantly higher expected maternal ages at first birth by approximately 11.2%, 4.2%, and 4.0%, respectively ($p < 0.01$). These estimates indicate comparatively later initiation of childbearing among women in these regions within the observed sample.

Conversely, women who had gave birth in Amhara, Oromia, and Gambela exhibited significantly lower expected maternal ages at first birth by approximately 4.4%, 3.0%, and 7.8%, respectively ($p < 0.05$), indicating earlier child-bearing relative to the reference region. The magnitude of the association was largest for Gambela. For SNNPR and Benishangul-Gumuz, the estimated reductions in maternal age at first birth were approximately 2.7%, with statistical significance close to conventional thresholds ($p \approx 0.05$), suggesting weaker evidence of earlier childbearing relative to Tigray. In contrast, the estimated effects for Harari and Afar were small and statistically insignificant, indicating no clear difference in expected maternal age at first birth compared with the reference region. Overall, these findings suggest substantial geographic heterogeneity in maternal age at first birth among women who had already experienced a first birth in Ethiopia.

Religion was also associated with variation in maternal age at first birth. Compared with Muslim women (reference category), Orthodox Christian women had a significantly higher expected maternal age at first birth of approximately 4.4% $(\exp(0.043) - 1) \times 100\%$, $p < 0.001$. Similarly, Protestant women exhibited an expected maternal age at first birth approximately 3% higher than that of Muslim women.

In contrast, women belonging to other religious groups (including Catholic, traditional beliefs, and other affiliations) had an expected maternal age at first birth approximately 5% lower than that of Muslim women, and this difference was statistically significant. These results indicate that maternal age at first birth varies across religious groups within the sample of women who had already experienced childbirth.

After controlling for region and religion, women who had already experienced a first birth and were residing in rural areas had a significantly lower expected maternal age at first birth compared with their urban counterparts. Specifically, the estimated coefficient ($\beta = -0.046$) corresponds to an approximately 4.5% lower expected age at first birth for rural women relative to urban women $[1 - \exp(-0.046)] \times 100\% = 4.5\%, p < 0.001[1 - \exp(-0.046)] \times 100\% = 4.5\%, p < 0.001[1 - \exp(-0.046)] \times 100\% = 4.5\%, p < 0.001$. This result indicates that, within the study sample of women who had already given birth, rural residence is associated with comparatively earlier initiation of childbearing than urban residence.

In a Gamma regression, the shape parameter is the inverse of the dispersion (variability). A higher shape implies lower dispersion (more homogeneous maternal age at first birth), and a lower shape implies higher dispersion (greater heterogeneity). Negative coefficients thus imply greater variability in maternal age at first birth compared to the reference category.

The dispersion sub-model examines factors associated with variability in maternal age at first birth among women who had already experienced a first birth. In this parameterization, negative coefficients correspond to lower shape parameters, which indicate greater dispersion (higher variability) in the age at first birth relative to the reference category.

Across regions, most estimated coefficients are negative and statistically significant, suggesting that maternal age at first birth is more variable in many regions compared with the reference region, Tigray. Women who had already experienced a first birth in Afar ($\beta = -0.26$) exhibited a substantially lower shape parameter, indicating higher variability in maternal age at first birth. Similarly, Benishangul-Gumuz ($\beta = -0.32$) and Harari ($\beta = -0.22$) show relatively large negative coefficients, suggesting greater heterogeneity in the timing of first birth among women in these regions.

Moderate but statistically significant increases in variability were also observed in Amhara ($\beta = -0.18$), Oromia ($\beta = -0.20$), and SNNPR ($\beta = -0.24$), indicating wider dispersion in maternal age at first birth relative to Tigray. In contrast, Addis Ababa ($\beta = 0.24$) and Dire Dawa ($\beta = 0.14$) exhibited positive coefficients in the dispersion model, corresponding to

higher shape parameters and therefore lower variability in maternal age at first birth compared with the reference region. This suggests that the timing of first birth among women in these regions is relatively more consistent within the observed sample.

Regarding religion, Orthodox Christian women exhibited significantly greater variability in maternal age at first birth compared with Muslim women (reference category) ($\beta = -0.150$, p = 0.008). This negative coefficient indicates a lower shape parameter and therefore greater dispersion in the age at first birth within this group. In contrast, the dispersion estimates for Protestant women and those belonging to other religious groups were not statistically different from the Muslim reference category, suggesting similar levels of variability in maternal age at first birth.

After accounting for regional and religious differences, rural women who had already experienced a first birth exhibited approximately 7% greater variability in maternal age at first birth compared with urban women. This result indicates that the timing of first birth among rural women in the sample is more dispersed than among their urban counterparts.

Overall, the dispersion model highlights notable differences in the consistency of maternal age at first birth across regions, religions, and place of residence, suggesting that variability in the timing of first childbirth differs across socio-demographic contexts within Ethiopia.

## Bayesian Gamma regression model

To support the classical estimation, Bayesian inference using MCMC was conducted for GLM GLMM and distributional gamma regression. Proper Bayesian diagnostics are required to ensure convergence, model reliability, and posterior accuracy (Gelman, 2013). The potential scale reduction factor (R) was employed to assess convergence across chains. As shown in Table 7, all R values were below the threshold of 1.01, confirming convergence for all model parameters.

**Table 7. Posterior Summaries and Convergence Diagnostics from Bayesian Gamma Regression Models (GLM and GLMM) for Maternal Age at First Birth.**

| Variable | Categories | GLM | | | | GLMM | | | |
|---|---|---|---|---|---|---|---|---|---|
| | | Estimate | Mcse | $\hat{R}$ | $n_{eff}$ | Estimate | Mcse | $\hat{R}$ | $n_{eff}$ |
| Intercept | | 2.93047 | 0.00039 | 1.00223 | 1473 | 2.92856 | 0.00026 | 1.00075 | 3266 |
| Residence | Urban | | | | | | | | |
| | Rural | −0.046 | 0.00011 | 1.00008 | 5232 | −0.0462 | 0.00008 | 0.99971 | 10271 |
| Religion | Muslim | | | | | | | | |
| | Orthodox | 0.042 | 0.00016 | 1.00159 | 2918 | 0.04024 | 0.00011 | 0.99996 | 6079 |
| | Protestant | 0.035 | 0.00016 0.00030 | 1.00038 | 4208 | 0.03478 | 0.00011 | 0.99990 | 9507 |
| | Others | −0.053 | | 1.0000 | 5294 | −0.0778 | 0.00021 | 1.00012 | 10495 |
| Region | Tigray | | | | | | | | |
| | Afar | −0.037 | 0.00039 | 1.00219 | 1793 | −0.0574 | 0.00026 | 1.00013 | 3752 |
| | Amhara | −0.0467 | 0.00027 | 1.00054 | 2303 | −0.05119 | 0.00019 | 1.00107 | 4957 |
| | Oromia | −0.0322 | 0.00034 | 1.00236 | 1809 | −0.03649 | 0.00023 | 1.00042 | 3722 |
| | Somali | 0.0935 | 0.00038 | 1.00191 | 1985 | 0.03583 | 0.00026 | 1.00017 | 4117 |
| | SNNPR | −0.0284 | 0.00033 | 1.00216 | 1977 | −0.03289 | 0.00022 | 1.0003 | 4250 |
| | Benishangul | −0.0318 | 0.00034 | 1.00293 | 1944 | −0.03841 | 0.00022 | 0.99995 | 4578 |
| | Gambela | −0.0843 | 0.00034 | 1.00248 | 2071 | −0.08705 | 0.00024 | 1.00028 | 4204 |
| | Hareri | 0.0164 | 0.00037 | 1.00136 | 1861 | 0.01046 | 0.00024 | 1.00022 | 4092 |
| | Addis Ababa | 0.10594 | 0.00035 | 1.00138 | 2171 | 0.10100 | 0.00025 | 0.99999 | 4483 |
| | Dere-dewa | 0.03868 | 0.00037 | 1.00188 | 1875 | 0.03429 | 0.00025 | 1.00037 | 3913 |
| Variation | Shape | 20.7452 | 0.00452 | 1.00005 | 7306 | 11.4501 | 0.00102 | 1.01000 | 3451 |

Visual inspection of trace plots (Fig 5) for parameters such as the intercept and Afar region further supports this conclusion. Each chain displayed overlapping paths with no systematic trends, indicative of successful exploration of the posterior distribution.

Another alternative way to assess the fit of the Bayesian models, posterior predictive check was conducted. This approach involves generating replicated datasets from the posterior predictive distribution and comparing these simulations to the observed data [43]. The posterior predictive check (PPC) of the mean age at first birth (AFB) demonstrates that the model's simulated datasets closely reproduce the distribution observed in the empirical data. The black curve, in Fig 6 representing the observed mean AFB distribution, peaks around 17–18 years and exhibits a gradual decline toward higher ages. The green curves, representing the posterior predictive simulations, overlap almost entirely with the observed distribution, indicating that the model accurately captures the central tendency, shape, and spread of the data. This strong alignment suggests that the Bayesian model provides an adequate fit to the observed mean AFB values, with no substantial bias or distortion in its predictions.

The agreement between the observed and simulated curves in Fig 6 across the entire distribution, including both tails, highlights the model's capacity to reflect the variability present in the data. The close correspondence implies that the prior specification, likelihood choice, and posterior estimates together provide a realistic generative mechanism for the observed patterns of AFB. Consequently, this PPC offers strong evidence of model adequacy in representing the key

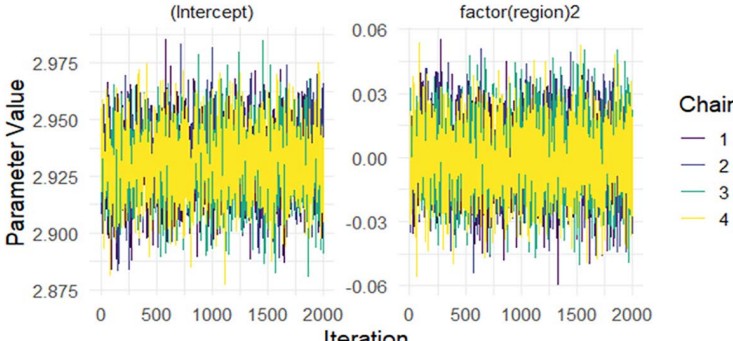

**Fig 5. Post-warmup trace plot for the slope on intercept and (region) 2 (Afar).**

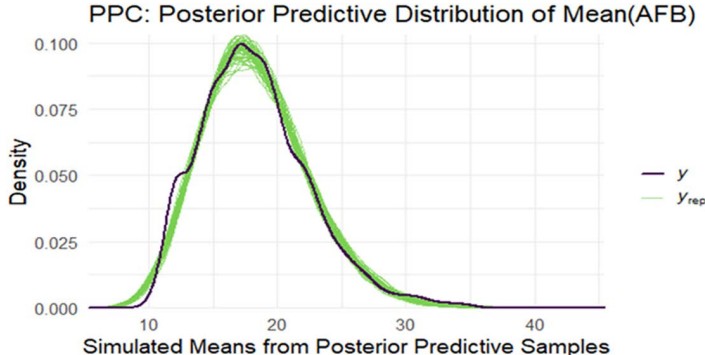

**Fig 6. Graphical posterior predictive check for posterior predictive distribution of AFB.**

distributional characteristics of mean AFB, reinforcing the validity of subsequent inferences and policy-relevant conclusions drawn from the model outputs.

The other numerical critical diagnostic is the effective sample size (ESS) or $n_{eff}$, which reflects the number of independent samples equivalent in information content to the correlated MCMC draws. Table 7 shows that ESS values exceed 1000 for all estimated parameters in models, satisfying standard convergence and precision criteria for Bayesian analysis [43].

The Monte Carlo standard error (MCSE) provides insight into the accuracy of MCMC approximations. A low MCSE relative to the posterior standard deviation indicates reliable posterior estimation. In Table 7 both GLM and GLMM models, all parameter MCSE values were minimal and well below the benchmark of 0.1, affirming that the posterior estimates are precise and the sampling process efficient [44].

The $\hat{R}$ (Gelman–Rubin diagnostic) values reported in Table 7 provide evidence of convergence for the Bayesian Gamma regression models under both GLM and GLMM frameworks. A $\hat{R}$ value close to 1.0 indicates that the Markov Chain Monte Carlo (MCMC) chains have mixed well and converged to the target posterior distribution. In these results, all covariates show R values ranging from 0.999 to 1.002, which is well within the commonly accepted threshold of <1.01, confirming excellent convergence across parameters. Even the shape parameter in the GLMM model, with R = 1.01, remains within acceptable limits, suggesting only a minimal deviation but still indicative of reliable convergence. Overall, these diagnostics demonstrate that the Bayesian estimation process was stable and produced trustworthy posterior summaries for both GLM and GLMM models.

As shown in Table 3, the distributional Gamma regression model estimated under the Bayesian framework demonstrated better predictive performance compared with the standard Gamma Generalized Linear Model (GLM) and the Gamma Generalized Linear Mixed Model (GLMM). This model jointly estimates both the mean (location) and the dispersion (shape) components of maternal age at first birth (AFB). The mean sub-model, specified with a log-link function, estimates the expected age at first birth as a function of the explanatory variables, while the dispersion sub-model captures differences in variability across regions, religions, and residential settings (Table 8).

Convergence diagnostics indicate that the Bayesian estimation performed well. All R values are equal to 1.00, indicating excellent convergence of the Markov Chain Monte Carlo (MCMC) simulations. In addition, the Bulk Effective Sample Size (Bulk_ESS) and Tail Effective Sample Size (Tail_ESS) values are sufficiently large, suggesting stable and reliable posterior estimates for all parameters.

The log-link function ensures that predicted values for both the mean and the shape parameters remain positive. Bayesian estimation was implemented using the No-U-Turn Sampler (NUTS) algorithm, which provides robust posterior sampling and credible interval estimation. The intercept estimate for the mean sub-model (β = 2.93) represents the expected log maternal age at first birth for the reference group defined as women residing in Tigray, belonging to the Muslim religion, and living in urban areas. After exponentiation, this corresponds to an expected maternal age at first birth of approximately 18.7 years (exp(2.93)), with a 95% credible interval of [18.2, 19.3] years. In the dispersion sub-model, the intercept estimate (β = 3.53) represents the baseline shape parameter, indicating relatively low variability in maternal age at first birth for the reference group.

The results presented in Table 8 indicate notable regional variation in maternal age at first birth among women who had already experienced a first birth, after adjusting for religion and place of residence. Relative to Tigray, women residing in Addis Ababa had an estimated 11.6% higher expected age at first birth (95% credible interval: 7.3%–15%), while those in Dire Dawa had approximately 4% higher expected age (95% credible interval: 1%–7.3%). These positive associations indicate comparatively later initiation of childbearing among women in these regions within the observed sample.

In contrast, women in several regions exhibited significantly lower expected ages at first birth relative to Tigray. For example, women residing in Gambela had an estimated 7.7% lower expected age at first birth (95% credible interval: 5%–10%). Similarly, reductions in expected age at first birth were observed in Amhara (4%), Oromia (3%), SNNPR

**Table 8. Bayesian Gamma Regression Estimates of Mean and Dispersion Parameters for Maternal Age at First Birth by Region, Religion, and Residence in Ethiopia.**

| | Estimate | Est.Error | l-95% u-95% CI | $\hat{R}$ | Bulk_ESS Tail_ESS |
|---|---|---|---|---|---|
| Intercept | 2.93 | 0.01 | 2.90 2.96 | 1.00 | 3765 5879 |
| Shape inter | 3.53 | 0.09 | 3.34 3.71 | 1.00 | 3330 5384 |
| Afar | 0.0021 | 0.02 | −0.03 0.03 | 1.00 | 4670 7326 |
| Amhara | −0.04 | 0.01 | −0.07 −0.02 | 1.00 | 6723 8552 |
| Oromia | −0.03 | 0.01 | −0.06 −0.01 | 1.00 | 4565 6653 |
| Somali | 0.04 | 0.02 | 0.01 0.07 | 1.00 | 4633 6862 |
| SNNPR | −0.03 | 0.01 | −0.05 −0.01 | 1.00 | 5346 7809 |
| Benishangul | −0.03 | 0.01 | −0.06 −0.01 | 1.00 | 5143 7627 |
| Gambela | −0.08 | 0.01 | −0.11 −0.05 | 1.00 | 5003 7074 |
| Hareri | 0.02 | 0.02 | −0.01 0.05 | 1.00 | 4897 7352 |
| Addis Ababa | 0.11 | 0.02 | 0.07 0.14 | 1.00 | 6573 8820 |
| Dere-dawa | 0.04 | 0.01 | 0.01 0.07 | 1.00 | 4617 7464 |
| Orthodox | 0.04 | 0.01 | 0.03 0.06 | 1.00 | 6160 8090 |
| Protestant | 0.03 | 0.01 | 0.02 0.05 | 1.00 | 9903 9257 |
| Others | −0.05 | 0.02 | −0.07 −0.02 | 1.00 | 14448 9160 |
| Rural | −0.05 | 0.01 | −0.06 −0.03 | 1.00 | 15367 9645 |
| **Shape (the inverse of dispersion)** | | | | | |
| Afar | −0.26 0.10 | | −0.33 −0.19 | 1.00 | 4045 6132 |
| Amhara | −0.18 0.08 | | −0.35 −0.02 | 1.00 | 5006 7641 |
| Oromia | −0.20 0.09 | | −0.37 −0.04 | 1.00 | 3791 5734 |
| Somali | −0.31 0.11 | | −0.52 −0.10 | 1.00 | 4187 6574 |
| SNNPR | −0.24 0.09 | | −0.33 −0.15 | 1.00 | 4312 6682 |
| Benishangul | −0.32 0.10 | | −0.51 −0.13 | 1.00 | 4134 6701 |
| Gambela | −0.28 0.10 | | −0.47 −0.08 | 1.00 | 4507 7633 |
| Hareri | −0.23 0.10 | | −0.42 −0.04 | 1.00 | 4188 7020 |
| Addis Ababa | 0.25 0.11 | | 0.04 0.46 | 1.00 | 5363 7433 |
| Dere-dawa | 0.14 0.10 | | 0.05 0.33 | 1.00 | 4334 6774 |
| Orthodox | −0.15 0.06 | | −0.26 −0.04 | 1.00 | 6297 8206 |
| Protestant | −0.08 0.07 | | −0.21 0.04 | 1.00 | 9826 9644 |
| Others | −0.17 0.14 | | −0.44 0.10 | 1.00 | 14005 9463 |
| Rural | −0.07 0.05 | | −0.09 −0.04 | 1.00 | 12020 9763 |

(2.9%), and Benishangul-Gumuz (3%), with credible intervals that did not include zero, indicating statistically meaningful differences relative to the reference region. Overall, these estimates suggest substantial geographic heterogeneity in maternal age at first birth among women who had already experienced childbirth.

Religion was also associated with differences in the expected age at first birth. After adjusting for region and place of residence, women identifying as Orthodox Christian had an expected maternal age at first birth approximately 4% higher than that of Muslim women (β = 0.04; 95% credible interval: 0.03–0.06). Similarly, Protestant women had an expected age at first birth approximately 3% higher than that of Muslim women (β = 0.03; 95% credible interval: 0.02–0.05).

In contrast, women categorized under other religions (including Catholic, traditional beliefs, and other affiliations) had an expected maternal age at first birth approximately 5% lower than that of Muslim women (β = −0.05; 95% credible

interval: −0.07 to −0.02). These results indicate differences in maternal age at first birth across religious groups within the study sample.

After adjusting for region and religion, rural residence was associated with a lower expected maternal age at first birth compared with urban residence. Women who had already experienced a first birth and resided in rural areas had an expected age at first birth approximately 5% lower than that of urban women (β = −0.05; 95% credible interval: −0.06 to −0.03). This indicates comparatively earlier initiation of childbearing among rural women within the sample.

In the case of precision (shape) or inverse dispersion sub-model region, religion and place of residence were associated with the maternal age at first birth variability. Positive coefficients for each covariate categories means, higher precision and lower dispersion (variability).

A tight way to interpret shape sub-model is changing shape results in to change in dispersion by the formula dispersion = $(\exp(-\beta_j)-1)\times100\%$. After adjusting residence and religion, most regions except Addis Ababa and Dere-Dewa show negative coefficients in the shape sub-model. Women who had already give birth in Somali have about 36.3% ($[e^{-(-0.31)} - 1]$ 100%) higher dispersion (variability), while those in Benishangul exhibit a 37.7% higher variability. Similarly, SNNPR (27.1%), Gambela (32.3%), and Harari (25.9%) also display greater heterogeneity in maternal age at first birth. This suggests that in these regions, women's age at first birth is much more diverse, with some starting childbearing very early and others delaying, reflecting socio-cultural and economic differences. In contrast, Addis Ababa and Dire Dawa show positive coefficients for the shape parameter, meaning they are associated with lower variability.

Specifically, Addis Ababa has about a 22.1% reduction in dispersion, while Dire Dawa shows a 13.1% reduction relative to the Tigray. This indicates that in urban and administrative centers, women's age at first birth tends to cluster more tightly around the mean, with fewer extremes of very early or very late first births. Such homogeneity could reflect greater access to education, reproductive health services, family planning and delay of early marriage. Amhara, Oromia, and Afar also show elevated variability, though to a more moderate degree. For instance, Amhara has a 19.7% higher dispersion; Oromia has a 22.1% higher dispersion, and Afar a 29.8% higher dispersion. These moderate increases suggest that, although the majority of women still enter childbearing at relatively young ages, there is growing variation, possibly reflecting changing norms among educated or urban subgroups compared to traditional practices in rural and pastoralist areas.

Religious affiliation also predicts variability. Relative to the Muslim reference group (assuming your coding), Orthodox Christians show a 16.2% higher dispersion and those identifying as other religions have a 18.5% higher dispersion, suggesting greater heterogeneity in the timing of first birth. Protestants, however, show a smaller and uncertain effect, with only an 8.3% increase in dispersion, and the 95% CrI crosses zero, indicating no strong evidence of a difference. These results align with prior demographic studies that find religious affiliation shapes fertility norms and the age of family formation differently across Ethiopian communities.

Rural women exhibit 7.3% greater dispersion in the age at first birth compared with urban women. This means that rural areas not only tend to have earlier mean ages at first birth but also a wider spread, reflecting more inequality in reproductive trajectories. While many rural women begin childbearing very early, a subset may delay slightly due to educational opportunities, regional migration, or access to health services, thereby widening the variability compared to more uniform urban experiences.

The primary rationale for employing three modeling frameworks GLM, GLMM, and distributional Gamma regression was to assess whether the associations between key covariates and maternal age at first birth remained consistent across statistical models of increasing flexibility. The results reported in Tables 4, 6–8 show that both Maximum Likelihood Estimation (MLE) and Bayesian estimation produced coefficient estimates that were broadly consistent in both magnitude and direction across the three modeling approaches. This agreement suggests that the observed relationships between region, religion, place of residence, and maternal age at first birth are relatively stable and not strongly dependent on the specific estimation framework used. Such consistency across estimation methods provides additional confidence in the robustness of the findings.

At the same time, moving from simpler models such as the GLM to more flexible approaches including the GLMM and distributional Gamma regression provided methodological advantages. The GLM offers a useful baseline for examining associations between predictors and the mean maternal age at first birth. The GLMM extends this framework by accounting for potential clustering and unobserved heterogeneity through random effects, which may improve estimation when observations are not fully independent. The distributional Gamma regression model further extends the analysis by allowing covariates to influence not only the mean of maternal age at first birth but also its dispersion. This approach provides a more comprehensive representation of the data structure by capturing both differences in expected age and differences in variability across socio-demographic groups. Consequently, although the substantive conclusions regarding the direction of covariate effects remain similar across models, the more flexible models provide additional insight into heterogeneity in the timing of first birth.

Taken together, the results from the distributional Gamma regression model provide a more detailed understanding of maternal age at first birth among women who had already experienced a first birth in Ethiopia. By jointly modeling the mean and dispersion of the outcome, this framework highlights not only which groups tend to have relatively earlier or later first births, but also where the timing of first birth is more or less variable across subpopulations.

The mean sub-model indicates that maternal age at first birth differs across regions, religious groups, and place of residence. In particular, women residing in urban areas and those living in regions such as Addis Ababa and Dire Dawa tend to exhibit higher expected ages at first birth compared with the reference region. Conversely, lower expected ages at first birth were observed in several regions including Gambela, Amhara, Oromia, SNNPR, and Benishangul-Gumuz. These results highlight geographic and socio-demographic differences in the timing of first childbirth within the study population.

The dispersion (shape) sub-model further reveals that variability in maternal age at first birth also differs across regions and residential settings. Several regions including Afar, Benishangul-Gumuz, Harari, Gambela, Somali, SNNPR, Oromia, and Amhara exhibit lower shape parameters relative to Tigray, indicating greater dispersion in the age at first birth among women in these areas. In contrast, regions such as Addis Ababa and Dire Dawa show relatively higher shape parameters, corresponding to lower variability in the timing of first birth within the observed sample.

Differences in variability were also observed across religious groups. While Orthodox Christian women showed higher expected maternal age at first birth in the mean model, the dispersion model indicates greater variability in the timing of first birth within this group relative to Muslim women. In contrast, Protestant women and those belonging to other religious groups showed no statistically meaningful differences in variability compared with the reference category.

Place of residence also showed consistent associations with both the expected age and variability of first birth. Rural women who had already experienced a first birth exhibited lower expected ages at first birth compared with urban women, and the dispersion model indicates greater variability in maternal age at first birth among rural residents. These results suggest that both the timing and the consistency of first birth differ between rural and urban populations within the sample.

Overall, the distributional Gamma regression framework provides a comprehensive perspective by simultaneously examining differences in both the average timing and the variability of maternal age at first birth. This dual modeling approach helps characterize heterogeneity in reproductive timing across regions, religions, and residential settings among Ethiopian women who had already experienced childbirth.

## Discussion

Maternal age at first birth is a critical indicator of reproductive health, population dynamics, and the socio-economic well-being of a society [2,3]. Evidence-based strategies to delay the timing of first birth can substantially reduce risks to both maternal and child health [4]. Although Ethiopia has made notable progress in increasing maternal age at first birth, considerable disparities persist across regions and religious groups among women who have already given birth [10,14,27]. Such inconsistencies hinder efforts to improve health outcomes for mothers and children [5,9,10]. This study investigates the magnitude and determinants of maternal age at first birth in Ethiopia, drawing on data from 5,839 women

included in the 2019 Ethiopia Mini Demographic and Health Survey (Mini EDHS), conducted between March 21 and June 28, 2019.

This study utilizes a robust comparative modeling framework to examine maternal age at first birth (AFB). Given the right-skewed distribution of AFB (ranging from 10 to 49 years), the Gamma distribution was deemed most appropriate [37]. Specifically, we fitted Generalized Linear Models (GLM), Generalized Linear Mixed Models (GLMM), and distributional Gamma regression models under both classical (Maximum Likelihood Estimation) and Bayesian estimation via MCMC. This multi-model approach provided a comprehensive assessment of both the mean and dispersion of AFB, thereby capturing disparities that are often obscured when focusing solely on central tendencies.

From a methodological perspective, the results demonstrated that distributional Gamma regression consistently outperformed the standard GLM and GLMM under both estimation frameworks. The modeling both the shape and location parameters, were improving model fit, as evidenced by lower AIC, BIC, and deviance values (under MLE), and higher expected log predictive density (ELPD) using Bayesian leave-one-out (LOO) cross-validation. Unlike standard models, it jointly estimated the expected value (mean) and the variance (via the shape parameter), offering richer insights into both the timing and variability of childbearing. Bayesian implementation further enhanced inference quality, with excellent convergence diagnostics (R ≈ 1.00, high Bulk_ESS and Tail_ESS, and low MCSE), confirming robust posterior stability and credibility.

The findings reveal that the average age at first birth (AFB) in Ethiopia remains around 18 years with credible intervals ranging 18.2 to 19.3. This statistic aligns broader trends observed in the literature, where significant proportions of motherhood in Ethiopia fail to meet the recommendation of a minimum of 18 years during maternity [9,19]. Means that, a substantial proportion of births still occur below both the WHO-recommended minimum of 18 years and the sub-Saharan African average of 19 years [5]. This underscores a persistent public health concern, as early childbearing increases risks of maternal morbidity, poor neonatal outcomes, and intergenerational cycles of poverty [45]. Moreover, the average is far below the internationally recognized optimal window for first birth (late 20s to early 30s), underscoring the urgency of interventions. This difference may be attributed to variations in study design, methodological approaches, the selection of contextual variables, and the focus on mothers who had already given birth. Our finding is consistent with previous studies conducted in Ethiopia, which reported mean ages at first birth of 20 years [29] and 19 years [28], as well as with evidence from Bangladesh (16.34 years) [46], Kenya (20.3 years) [47], Swaziland (18.22 years) [48], Nigeria (19 years) [21], and Uganda (19.2 years) [49]. This similarity may be explained by the high prevalence of early marriage and initiation of sexual activity in these countries [29]. Early marriage often limits women's autonomy in reproductive health decisions, resulting in early childbearing [50]. Another contributing factor could be the restricted educational opportunities for girls, particularly in predominantly rural populations, which encourages early marriage as a means of obtaining social and financial support [51]. In contrast, our result was notably lower than that observed in developed countries, where the mean age at first birth exceeds 30 years [51]. This difference may be attributed to the fact that adolescent girls in developed contexts are more likely to remain in school and many women enter the workforce to gain economic independence, thereby postponing childbearing [52,53]. Furthermore, women in developed countries generally possess better awareness of the risks associated with early childbirth and greater access to family planning services, which supports delaying the first birth [54].

While the national average underscores a persistent public health concern, further insights emerge when examining key factors such as region, religion, and residence. Although these variables were measured at the time of the survey, they are reasonably assumed to have remained stable over the preceding five years, thereby serving as valid determinants of maternal age at first birth. These factors consistently shape both the mean and variability of AFB across models. Marked regional disparities were observed. Compared with Tigray, women who had already give birth in Amhara (4%), Oromia (3%), SNNPR (3%), Benishangul-Gumuz (3%), and Gambela (8%) tended to initiate first births at younger ages, whereas those in Addis Ababa (11%), Somali (4%), and Dire Dawa (4%) delayed their first births. These patterns reflect underlying inequalities in education, economic development, and access to health services. This statistics were consistent

with findings from previous studies in Ethiopia and sub-Saharan Africa [8,32,45–47]. Notably, the distributional model revealed that regions such as Afar (37.7%), Benishangul-Gumuz (37.7%), Somali (36%), Gambela (32.3%), SNNPR (27.1%), and Harari (25.9%) exhibited higher variability in maternal age at first birth. This suggests heterogeneous fertility norms and unequal access to reproductive health services across these regions. In contrast, Addis Ababa (22%) and Dire Dawa (13%) not only had later average ages at first birth but also lower variability, reflecting more uniform reproductive patterns likely influenced by higher education levels, urbanization, and stronger health system infrastructure. These findings highlight important regional disparities with implications for targeted maternal and child health interventions.

Religion also associated in shaping AFB. Muslim women consistently had the lowest AFB, while Orthodox and Protestant women tended to delay childbirth. These results align with findings from Ethiopia [55], Uganda [49], and Nigeria [56], where religion strongly influences marriage timing, contraceptive uptake, and gender expectations. This is supported by other studies in Ethiopia and Nigeria in which early childbirth was found to be higher among Muslim followers than Orthodox [29,57]. The association between the Muslim religion and early childbirth might be attributed to the low uptake of contraceptive methods by Muslim followers [22,32]. Sometimes, some Muslims claim that having children is a source of blessing [9]. Besides, women's educational attendance is low in the eastern part of the country where Muslims predominate and this might contribute to early marriage and early childbirth [9]. The distributional Gamma regression further revealed that Orthodox women showed greater variability in AFB, possibly reflecting tensions between modernization (delayed marriage and career pursuits) and traditional norms of early family formation.

Rural women were more likely to begin childbearing earlier than their urban counterparts, a finding corroborated by prior Ethiopian [9,28,58] and sub-Saharan regional studies [47,59] studies. This disparity reflects systemic barriers in rural areas, including limited education, weaker health infrastructures, higher rates of early marriage, and entrenched traditional norms. Rural women who had already give birth, tended to have 5% early average ages at first birth compared to urban counterparts. The reason for this might be because those women residing in rural areas are less knowledgeable about early birth impact, have less accessible to health information and could not easily access and utilize family planning services compared with urban dwellers. The presence of low utilization of modern contraceptive methods might contribute to hastened first child birth in rural areas [60]. Rural residents might have a high interest in having many children and start giving birth early on time. Besides, early marriage might be high in those rural areas. Moreover, women in rural areas are less likely to be educated and less likely to be from educated parents, which means they have poor awareness of the consequences of early childbirth and a high unmet need for contraceptives [21,61]. The variability in timing was also higher compared to urbanized or mixed regions. This greater variability may reflect uneven access to education, reproductive health services, and family planning in rural areas, combined with diverse cultural norms around marriage and childbearing. Such disparities result in some women marrying and giving birth very early, while others delay due to schooling, migration, or socioeconomic differences.

While these associations highlight the impact of religion, region, and residence in shaping maternal age at first birth, it is important to acknowledge that such factors rarely act independently. Their effects are often mediated through socioeconomic pathways, including maternal education, marital practices, occupation, and access to health services, as documented in previous studies [9,28,31,51]. For instance, the earlier AFB observed among Muslim and rural women may partly reflect lower levels of educational attainment and earlier marriage, whereas the greater variability among Orthodox and rural populations could be linked to unequal access to family planning and diverse cultural norms [19,27,29]. To assess the extent of such confounding, we conducted a sensitivity analysis by incorporating key socioeconomic covariates. This approach enabled us to disentangle the extent to which observed disparities were attributable to socioeconomic inequalities versus deeper cultural and structural determinants that continue to shape reproductive behavior.

The sensitivity analysis further allowed us to assess potential bias from omitted socioeconomic variables. As shown in Table 5, the inclusion of maternal education, marital status, occupation, age at first marriage, and health service access reduced the coefficients for region, religion, and residence by less than ±10%. For instance, the effect of urban residence

decreased by 10.4%, while that of Addis Ababa dropped by 5.7%. Similarly, Muslim and Orthodox affiliation coefficients declined by 3.5% and 5.0%, respectively. These modest changes indicate that part of the observed disparities in maternal age at first birth can be attributed to differences in education, marriage timing, and health service availability. However, the persistence of statistically significant effects even after adjustment confirms that regional, religious, and residential differences reflect more than just socioeconomic inequalities; they also capture deep-rooted cultural norms, structural conditions, and geographic contexts that independently shape reproductive behavior of women who had already give birth in Ethiopia.

This study demonstrates that maternal age at first birth in Ethiopia remains alarmingly low, with profound disparities across regions, religions, and residential settings. By incorporating both mean and variability into the analysis, the distributional Gamma regression model provides a more nuanced understanding of these inequalities, offering critical evidence for equity-focused, context-specific interventions. Addressing both the timing and consistency of childbearing will be essential for achieving maternal and child health goals, reducing intergenerational cycles of disadvantage, and meeting Ethiopia's commitments to the Sustainable Development Goals.

## Strength and limitation of the study

The strengths include reliance on nationwide data and the employment of advanced statistical techniques in both mean and dispersion of maternal age at first birth, as well as adequate models which enhance the confidential applicability and reliability of the results. Moreover, it is believed that the Demographic and Health Survey (DHS), a widely accepted and validated instrument with a large, carefully curated sample size, further strengthens the credibility and accuracy of the results of this study.

This study has several limitations the authors need to acknowledge. First, it focused only on women who had given birth, excluding censored data, which may bias results and limit generalizability. Second, reliance on self-reported DHS data may introduce recall or reporting errors, especially for age at first birth and reproductive behaviors. Third, the cross-sectional design precludes causal inference between covariates and maternal age at first birth. Fourth, unmeasured confounders including maternal education, occupation, early marriage, marital status, household income, healthcare access and quality, cultural practices, and reproductive health knowledge may influence results. Fifth, regional and rural-urban classifications may mask intra-region heterogeneity, particularly in mobile or pastoralist populations, and age reporting errors may affect estimates of mean and variability. Finally, while the distributional Gamma regression captures mean and dispersion effectively, its parametric assumptions may not fully reflect extreme values or nonlinear patterns. Longitudinal studies incorporating broader socio-economic and cultural factors are warranted to strengthen causal inference and inform policy.

## Conclusion

This study analyzed maternal age at first birth (AFB) in Ethiopia using data from 5,839 women captured in the Mini EDHS 2019. The analysis focused exclusively on women who had experienced at least one live birth, ensuring precise insights by excluding censored observations. The average AFB was found to be 18.7 years, with values ranging from 10 to 40 years highlighting wide demographic diversity and underscoring the urgency of addressing early childbearing in specific subpopulations.

Given the positive, right-skewed, and non-negative nature of the AFB, Gamma regression was the most appropriate modeling choice. Its distributional properties align well with the structure of the data, enabling effective modeling of both the mean and variability of AFB. The analysis employed three modeling frameworks Generalized Linear Models (GLM), Generalized Linear Mixed Models (GLMM), and distributional Gamma regression each estimated using both classical (MLE) and Bayesian methods. Among these, the distributional Gamma models consistently outperformed the standard GLM and GLMM, as indicated by lower AIC, BIC, and deviance (under MLE) and higher Bayesian model fit scores such

as LOO, WAIC, the expected log predictive density (elpd). Bayesian models further showed strong convergence and improved parameter precision.

Across all models and estimation approaches, consistent and significant associations were identified between AFB and religious, regional and residential covariates. Women from rural areas, Muslim backgrounds, and regions like Amhara, Oromia, Benishangul-Gumuz, SNNPR and Gambela were more likely to experience early childbearing. In contrast, higher average AFB was observed among women in urban areas, particularly in regions such as Addis Ababa, Dire-Dawa and Somali, and among followers of Orthodox and Protestant religions. Importantly, the distributional Gamma model revealed that AFB variability itself is not constant greater dispersion was observed in certain regions like Afar, Gambela, Benishangul Gumuz, Amhara, Oromia, SNNPR, Somali and Harari and among Orthodox women, highlighting heterogeneous reproductive patterns within each group. Women who had already give birth in rural part of the country also have high variability in age at their first birth.

The findings provide robust evidence of geographical, religious, and residential disparities in reproductive average maternal age across Ethiopia. These insights point to the need for targeted policy interventions that are context-specific and equity-focused. Strategies should prioritize improving girls' education, delaying early marriage, and expanding access to culturally sensitive reproductive health services, particularly in rural and high-risk regions.

The average maternal age at first birth in Ethiopia remains at 18 years just at the lower limit of the World Health Organization's recommendation. Alarmingly, in certain regions, the age of first childbirth can be as young as 10 years. On the other hand, some women in Addis Ababa and Amhara delay childbirth until after age 35, indicating significant demographic diversity. This wide range not only emphasizes the urgency of intervention but also highlights the importance of contextualized, locally adapted solutions.

Therefore, policymakers should prioritize targeted programs that address the sociocultural and geographic determinants of early childbirth. Investments in female education, access to reproductive health services, and culturally sensitive community engagement strategies are essential. The dual perspective offered by the distributional model on both mean and variability has significant policy implications. First, programs must go beyond raising mean AFB and also address disparities in timing, since high variability often reflects inequities in access to education, health services, family planning and cultural autonomy. Second, interventions should be region-specific: for example, urban centers like Addis Ababa may require strategies targeting heterogeneity within mixed populations, while peripheral regions like Afar, Somali, Gambela and Benishangul need tailored programs that address pastoralist lifestyles, limited service coverage, and entrenched fertility norms. Third, religious institutions and leaders should be engaged as key partners, given the strong role of faith in shaping fertility practices. Fourth, rural focused policies must prioritize delaying age at marriage, expanding girls' secondary and higher education, strengthening youth-friendly reproductive health services, and addressing structural barriers to autonomy.

## Supporting information

**S1 Data. AFB data.**
(CSV)

## Author contributions

**Conceptualization:** Adimias Wendimagegn Agegnehu, Butte Gotu Arero.

**Data curation:** Adimias Wendimagegn Agegnehu, Butte Gotu Arero.

**Formal analysis:** Adimias Wendimagegn Agegnehu, Butte Gotu Arero.

**Funding acquisition:** Adimias Wendimagegn Agegnehu.

**Investigation:** Adimias Wendimagegn Agegnehu, Butte Gotu Arero.

**Methodology:** Adimias Wendimagegn Agegnehu, Butte Gotu Arero.

**Project administration:** Adimias Wendimagegn Agegnehu, Butte Gotu Arero.

**Resources:** Adimias Wendimagegn Agegnehu, Butte Gotu Arero.

**Software:** Adimias Wendimagegn Agegnehu, Butte Gotu Arero.

**Supervision:** Adimias Wendimagegn Agegnehu, Butte Gotu Arero.

**Validation:** Adimias Wendimagegn Agegnehu, Butte Gotu Arero.

**Visualization:** Adimias Wendimagegn Agegnehu, Butte Gotu Arero.

**Writing – original draft:** Adimias Wendimagegn Agegnehu, Butte Gotu Arero.

**Writing – review & editing:** Adimias Wendimagegn Agegnehu, Butte Gotu Arero.

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
