## [Decision Letter · Decision Letter 0]

11 Aug 2025

Dear Dr.  Agegnehu,

Thank you for submitting your manuscript to PLOS ONE. After careful consideration, we feel that it has merit but does not fully meet PLOS ONE’s publication criteria as it currently stands. Therefore, we invite you to submit a revised version of the manuscript that addresses the points raised during the review process.

We look forward to receiving your revised manuscript.

Kind regards,

Denekew Bitew Belay, Ph.D

Academic Editor

PLOS ONE

Journal Requirements:

Reviewer's Responses to Questions

**Comments to the Author**

1. Is the manuscript technically sound, and do the data support the conclusions?

Reviewer #1: Yes

Reviewer #2: Partly

Reviewer #3: Yes

Reviewer #4: Partly

2. Has the statistical analysis been performed appropriately and rigorously?

Reviewer #1: Yes

Reviewer #2: Yes

Reviewer #3: No

Reviewer #4: Yes

3. Have the authors made all data underlying the findings in their manuscript fully available?

Reviewer #1: Yes

Reviewer #2: Yes

Reviewer #3: Yes

Reviewer #4: Yes

4. Is the manuscript presented in an intelligible fashion and written in standard English?

Reviewer #1: Yes

Reviewer #2: No

Reviewer #3: No

Reviewer #4: Yes

Reviewer #1: Peer Review Report

Manuscript: PONE-D-25-07127

Title: Modelling Determinants of Maternal Age at First Birth in Ethiopia: Gamma Regression Approach

Dear Editor,

I have carefully reviewed the manuscript "Modelling Determinants of Maternal Age at First Birth in Ethiopia: Gamma Regression Approach." Below is my assessment of the manuscript's limitations and recommendations for improvement.

Major Concerns

Variable selection rationale: The authors' decision to exclude key socioeconomic variables severely limits the study's explanatory power. On page 14, the authors state: "The exclusion criteria of variables from this study were established to ensure that the analysis focused on factors that directly relate to the first childbearing status of mothers. Maternal characteristics such as educational level, occupation status, and marital status, while important in broader contexts, were deemed insufficiently relevant to the specific inquiry regarding maternal age at first childbirth." This justification contradicts substantial literature demonstrating that education and marital status are among the strongest predictors of maternal age at first birth. For example, Essey Kebede Muluneh et al. (2023), which the authors cite, found significant associations between these variables and age at first birth. By excluding these variables, the authors have created models that likely suffer from omitted variable bias. The authors should either: (1) include these critical variables in their models while acknowledging the temporal limitation, or (2) conduct sensitivity analyses to demonstrate that their exclusion does not substantially alter the findings related to region, religion, and residence.

Statistical modeling decisions: The manuscript presents gamma regression as the appropriate choice but provides insufficient evidence for this selection. On page 15, the authors merely state: "It was observed that the dataset on the maternal age at first birth obtained from the Mini EDHS 2019 dataset follows a gamma distribution and consists of only positive values." This statement lacks statistical support. Figure 2 shows a visual comparison between normal and gamma distributions, but no formal goodness-of-fit tests are presented. The authors should:

Present results from statistical tests comparing the fit of different distributions (e.g., Akaike Information Criterion values for models using gamma, log-normal, and Weibull distributions)

Provide Q-Q plots comparing observed data against theoretical distributions

Justify the use of the log link function over alternative link functions for gamma regression

Focus on women who had given birth only: The authors' decision to exclude women without births fundamentally changes the research question and limits comparability with previous research. On page 15-16, they state: "However our concern for this study is mothers who had their first child birth only. Maternal age at first birth consists of non-negative values within this specific age range during the survey period." This approach introduces potential selection bias. Women who had not yet given birth by the survey date (particularly younger women) are systematically excluded, artificially truncating the distribution of the outcome variable. This could lead to biased estimates of the effects of predictors. For example, if highly educated urban women tend to delay childbirth beyond the survey period, their exclusion would underestimate the true effect of urban residence on maternal age at first birth. The authors should:

Explicitly discuss this limitation and its implications for interpreting results

Consider employing methods that account for right-censoring, such as survival analysis, and compare results with their current approach

At minimum, provide descriptive statistics comparing included and excluded women on key demographic characteristics

Causal inference limitations: Throughout the manuscript, the authors use language implying causal relationships despite the cross-sectional nature of the data. For instance, on page 33, they state: "The place of residence plays a crucial role in determining the age at which Ethiopian women have their first child, with women living in rural areas tending to give birth at a younger age than their urban counterparts." This phrasing suggests that place of residence determines age at first birth, when the data only support an association. Similar causal language appears when discussing religion and region. The manuscript should consistently use associational language (e.g., "associated with" rather than "plays a crucial role in determining") and explicitly acknowledge that unobserved confounders may explain some of the observed relationships.

Theoretical framework: The manuscript lacks a coherent theoretical framework explaining why region, religion, and residence would influence maternal age at first birth. The introduction provides general background on early childbearing but does not adequately theorize the specific mechanisms through which the selected variables might operate. For example, when discussing regional differences on page 33-34, the authors note differences in expected maternal age at first birth across regions but do not discuss why these differences exist (e.g., cultural norms, economic development, educational opportunities). A stronger theoretical framework would:

Propose specific mechanisms linking each predictor to the outcome

Discuss potential interactions between predictors (e.g., how religion might influence age at first birth differently in urban versus rural settings)

Identify potential mediating variables (which might include some of the excluded variables like education and marital status)

Minor Issues

Presentation of results:

Table 3 and Table 4 contain excessive numerical information that impedes readability. For example, Table 4 presents all parameter estimates with their standard errors, R-hat statistics, and effective sample sizes, resulting in an overwhelming amount of information.

Current format: Full tables with all diagnostic statistics for every parameter.

Recommendation: Split these tables into summary tables of key findings and move detailed diagnostics to supplementary materials. Additionally, consider using visual representations (e.g., coefficient plots) to display regression results instead of dense numerical tables.

Literature review organization:

The literature review (pages 7-10) currently presents studies chronologically rather than thematically, making it difficult to identify research gaps.

Current organization: "In East Asia and the Pacific, Bangladesh, Nigeria, Ghana, Sub-Saharan Africa, and Uganda, the median ages at first birth are 20.2, 16.34, 20, 19, and 19.2 years, respectively (Kennedy E, 2011; Karim MA, 2021; Fagbamigbe AF, 2016; Ida LA, Albert L, 2015; Negash WD, Asmamaw DB, 2022; Demographic Health survey, 2016)."

Recommendation: Reorganize by themes such as "Regional Variations in Age at First Birth," "Methodological Approaches to Studying First Birth Timing," and "Socioeconomic Determinants of Early Childbearing." This would help highlight specific gaps this study addresses.

Statistical terminology inconsistencies:

The manuscript uses "gamma regression model," "gamma family GLM," and "gamma family with log link" interchangeably throughout sections 3.2-3.4.

On page 17, the authors refer to both "shape parameter" and "dispersion parameter" for the same concept.

The term "mixed-effects" is used on page 15, while "random effects" is used on page 16 to describe the same model component.

Recommendation: Standardize terminology throughout the manuscript. For example, consistently use "gamma regression with log link" and select either "shape parameter" or "dispersion parameter" and maintain consistency.

Grammatical errors and typographical issues:

Page 8, line 2: "There is a clear associate between the age..." should be "There is a clear association between the age..."

Page 9, line 3: "...in a women's further education..." should be "...in a woman's further education..."

Page 15, line 12: "It is important to explore these alternative methods to ensure that the analysis of the age at first birth data is both accurate and meaningful. In this study, it was observed that the dataset on the maternal age at first birth obtained from the Mini EDHS 2019 dataset (excluding mothers without birth during survey period) follows a gamma distribution and consists of only positive values." - This paragraph contains redundancy and run-on sentences.

Throughout the manuscript: Inconsistent capitalization of "Mini EDHS" versus "mini EDHS"

Recommendation: Conduct a thorough grammar and spelling check. Consider simplifying complex sentences, particularly in the methodology section.

Figures quality and presentation:

Figure 1a: The histogram and density plot have poor resolution, with pixelated axis labels.

Figure 2: The legend overlaps with the plot, obscuring some data points.

Figure 4: The side-by-side comparison of GLM and GLMM posterior predictive checks lacks clear axis labels and sufficient resolution for publication.

Appendix 4: The trace plots are compressed and difficult to interpret with overlapping chain colors.

Recommendation: Recreate all figures using higher resolution settings (minimum 300 dpi). Ensure consistent font sizes across all figures (recommend 9-12pt). Position legends outside the plot area to avoid overlap with data. Add more descriptive axis labels that include units of measurement.

Specific Recommendations

Introduction:

Current: "The study focused on determining maternal age at first birth and investigating the factors influencing the age at first birth among women in Ethiopia with adequate model performance."

Improved: "This study aims to identify and quantify the influence of geographical, religious, and residential factors on maternal age at first birth in Ethiopia, while evaluating the adequacy of gamma regression models for analyzing this outcome."

Methods:

Current: "In this study, it was observed that the dataset on the maternal age at first birth obtained from the Mini EDHS 2019 dataset follows a gamma distribution and consists of only positive values."

Improved: "We evaluated the distributional properties of maternal age at first birth in the 2019 Mini EDHS dataset using goodness-of-fit tests and Q-Q plots. Results indicated that the gamma distribution provides an appropriate fit for this strictly positive, right-skewed outcome variable compared to alternative distributions (e.g., log-normal, Weibull)."

Results:

Current: "The result indicates that women in rural areas had a 5% lower expected age at first birth compared to women living in urban area."

Improved: "After adjusting for region and religion, women residing in rural areas had a 4.6% lower expected age at first birth compared to urban women (β = -0.047, 95% CI: [-0.061, -0.031])."

Discussion:

Current: "With the average age for women's first childbirth at 18 years, which is lower than sub-Saharan Africa age at first birth of 19 years this study recommends collaboration between federal and regional stakeholders including religious institutions to enforce legal age regulations for first births."

Improved: "Our findings indicate that the average age at first childbirth in Ethiopia (18 years) remains below both the WHO recommended minimum age of 18 years and the sub-Saharan African average of 19 years. These results suggest a need for targeted interventions that address regional and religious disparities in early childbearing, potentially through coordinated efforts between federal agencies, regional authorities, and religious institutions to promote awareness and enforcement of existing legal age regulations."

Conclusion

This manuscript addresses an important topic but requires substantial revisions before publication in PLOS ONE. The methodological approach needs stronger justification, the presentation of results should be improved for clarity, and various inconsistencies in terminology and writing need to be addressed.

Reviewer #2: Review Comments to the Author

Thank you for the opportunity to review this manuscript titled “Modelling Determinants of Maternal Age at First Birth in Ethiopia: Gamma Regression Approach.” The study investigates an important demographic and public health issue by analyzing determinants of maternal age at first birth using data from the 2019 Ethiopia Mini DHS. While the topic is highly relevant, and the modeling approach is technically appropriate, the manuscript requires significant improvements in clarity, interpretation, and methodological justification before it is ready for publication.

1. Scientific and Technical Soundness

The analysis is largely methodologically sound. The Gamma regression model is an appropriate choice for modeling positively skewed continuous data such as maternal age at first birth. The use of both Generalized Linear Models (GLM) and Generalized Linear Mixed Models (GLMM) is commendable, as is the inclusion of both maximum likelihood and Bayesian estimation techniques with appropriate diagnostic checks (R < 1.1, high ESS, low MCSE).

However, the justification for excluding key socio-demographic variables (e.g., education, marital status, occupation) due to timing of measurement is a concern. These are important predictors of maternal age at first birth and their exclusion may introduce bias. At the very least, a deeper discussion of potential omitted variable bias and its implications on results should be provided.

Additionally, a brief rationale for choosing Gamma regression over time-to-event models (e.g., Cox proportional hazards, AFT models) would strengthen the manuscript, particularly given the nature of the outcome (age at first birth).

2. Data and Availability

The manuscript complies with PLOS ONE’s data availability policy. The data are drawn from a publicly accessible, de-identified source—the 2019 Ethiopia Mini DHS via the Measure DHS program. The authors clearly state that all datasets are fully available without restriction, which meets the journal’s requirements. However, they should provide the exact link to the data access page and any relevant dataset identifiers to facilitate reproducibility.

3. Statistical Rigor and Interpretation

The statistical analysis is rigorous, with well-justified use of a Gamma distribution and appropriate model diagnostics. The dual estimation approach (MLE and Bayesian) is a notable strength and enhances confidence in the robustness of the findings. Diagnostic techniques such as posterior predictive checks, convergence monitoring, and model fit metrics are all appropriately used.

That said, the interpretation of model results is relatively limited. The discussion would benefit from more explanation of why particular regional or religious groups are associated with earlier or later age at first birth. Linking statistical results to sociocultural or health system factors in Ethiopia would give the analysis greater depth and relevance.

Also, while model fit statistics like AIC and BIC are mentioned, it would be helpful to clearly present them in a summary table and offer a more direct comparison of the two models (GLM vs. GLMM) to guide reader understanding.

4. Writing Quality and Clarity

This is a major weakness of the manuscript. The English language quality is not sufficient for publication in its current form. The paper contains frequent grammatical errors, awkward phrasing, incorrect syntax, and unclear sentence structures that significantly hinder readability. For example:

• “The teenage a woman is…” → should be “The younger a woman is…”

• “mor far” → should be “more distant”

• Phrases such as “This issue is linked to various negative impacts…” are vague/unclear and redundant.

The Introduction and Background sections are too long and repetitive. Several paragraphs reiterate the same facts about early childbirth and maternal health risks, often using slightly different wording. This redundancy should be minimized to improve focus and clarity.

The Bayesian estimation section is also overly technical for a general readership. Simplifying the language and breaking up long blocks of equations and dense paragraphs will improve accessibility.

A professional English-language editor or native speaker with expertise in scientific writing should be engaged to revise the manuscript thoroughly.

5. Figures, Tables, and Visual Aids

Some of the figures (e.g., Figure 1a–1c, Figure 2) are not clearly labeled or well integrated into the discussion. Each figure should be accompanied by a complete, self-contained caption and referenced explicitly in the text with interpretation. The density plots and boxplots are helpful, but their readability can be improved.

Additionally, a summary table of model fit statistics (AIC, BIC, LOO, etc.) and key coefficient estimates with credible/confidence intervals would strengthen the presentation.

6. Ethics and Compliance

The authors state that the study received appropriate ethical approval from the IRB of ICF, with data anonymized to prevent identification. This aligns with ethical standards for secondary data analysis.

No concerns were identified regarding dual publication, misconduct, or research ethics.

Conclusion and Recommendation

This study has the potential to make a valuable contribution to the literature on early childbirth and maternal health in Ethiopia. The modeling approach is sound, and the use of both classical and Bayesian estimation adds methodological richness.

However, major revisions are required before the manuscript is suitable for publication. These include:

• Substantial improvement of English language and clarity

• More comprehensive discussion of excluded variables and potential biases

• Stronger interpretation of statistical results in their social and policy context

• Clearer presentation of model comparison and fit statistics

I encourage the authors to revise and resubmit the manuscript with these improvements. Once addressed, this work could make a meaningful contribution to public health research and policy in Ethiopia and similar settings.

Reviewer #3: 1. General Comments

This manuscript investigates the relationship between longitudinal CD4 count trajectories and survival in HIV patients on ART in Ethiopia using joint modeling techniques. The topic is clinically important and relevant, especially in low-resource settings where treatment monitoring is critical.

The authors have applied appropriate statistical techniques and reported their findings clearly. However, certain elements—especially the necessity and justification for joint modeling—require clarification. Furthermore, improvements in model reporting, comparison with separate models, and explanation of assumptions and missing data handling would enhance the manuscript’s methodological strength and transparency.

2. Specific Comments

2.1. Justification for Joint Modeling

Joint models are most appropriate when there is informative dropout or when the longitudinal process (CD4 count) influences the risk of the event (death). While the manuscript reports a significant association parameter (α), it does not provide sufficient justification that joint modeling was required over separate models.

Please clarify:

Whether missing data in CD4 counts were non-random or informative.

How separate models (LMM and Cox) performed in comparison to the joint model. Including such results would demonstrate the added value of joint modeling.

2.2. Association Structure

The model appears to use a shared random intercept to link the longitudinal and survival processes.

Please explain:

Why the random intercept alone was chosen—was random slope variability tested and ruled out?

Whether other association structures (e.g., current value or slope) were considered.

2.3. Handling of Missing Data

The manuscript does not describe the extent or mechanism of missingness in CD4 measurements.

Please include:

A description of the pattern and proportion of missing CD4 data.

Whether missingness was tested (e.g., MNAR vs MAR) and whether any sensitivity analyses were done.

2.4. Time-Varying Covariates

It is not clear how time-varying factors (e.g., adherence, regimen change) were modeled—if at all.

Please clarify whether such variables were:

Modeled dynamically, updated at each measurement point.

Or treated as baseline values.

2.5. Model Fit and Assumption Checks

Please expand on:

How model assumptions (e.g., proportional hazards in the Cox model) were tested.

Model diagnostics for residuals and random effects.

Any convergence issues or estimation warnings, particularly for the joint model.

3. Minor Comments

-Specify which R packages were used (e.g., JM, JMbayes, joineR).

-Clarify the clinical interpretation of the α parameter—what does the magnitude imply for patient risk?

-Tables and figures are generally informative. However, adding:

-Individual CD4 trajectories or spaghetti plots.

-A side-by-side comparison of separate model vs. joint model results (even as supplementary material), would be useful.

Reviewer #4: The manuscript on modeling maternal age at first birth using Gamma regression (GLM and GLMM) presents an interesting and relevant study; however, it is currently prepared in a style that resembles a thesis rather than a journal article and does not fully align with the PLOS One formatting requirements. The methodology section on Gamma regression and estimation should be presented more concisely, focusing on the study-specific application, as it is an established method. Figures require higher resolution, improved labeling, and more informative captions. The authors should clearly demonstrate that model assumptions have been checked and met. Since the study uses Demographic and Health Survey (DHS) data, the authors are encouraged to consider resampling techniques to address potential sampling bias or imbalance in the dataset. Additionally, grammar and sentence structure need improvement to enhance clarity and readability.

.

Reviewer #1: No

Reviewer #2: **Yes:** Adekunle Fatai AdeoyeAdekunle Fatai AdeoyeAdekunle Fatai AdeoyeAdekunle Fatai Adeoye

Reviewer #3: **Yes:** Dr. Muhammad AasimDr. Muhammad AasimDr. Muhammad AasimDr. Muhammad Aasim

Reviewer #4: No

---

## [Author Response · Author response to Decision Letter 1]

11 Sep 2025

Reviewer 1: All comments were revised accordingly

We thank the reviewer for highlighting this point. Our study focused on maternal characteristics directly influencing first birth timing. To address this concern, we conducted a sensitivity analysis including education, marital status, and occupation (results section, Table 5). Results show that their inclusion small changes in magnitude and does not change direction in the associations between region, religion, and residence with the response variable maternal age at first birth. We also clarified the limitations of variable exclusion in the discussion (page 46 under strengths and limitations sub title).

We now provide formal goodness-of-fit assessments comparing Gamma, Normal, log-normal, and Weibull distributions, including AIC values and Q-Q plots (Table 2, Figure 5). Results indicate that the Gamma distribution with log link best fits the strictly positive, right-skewed outcome. The log link ensures positive predicted values and allows multiplicative interpretation of covariate effects, consistent with standard practice.

We acknowledge this limitation and the study objective is to see only mothers who had already give birth. While survival analysis could accommodate censoring, the objective of this study is to see only women who had given birth and associations with regional religious and residential factors; thus, our approach remains appropriate for cross-sectional analysis. Results are interpreted as associations among women who have already had first birth.

All statements throughout the manuscript have been revised to use associational language (e.g., associated with instead of “determines”) and to acknowledge potential unobserved confounders.

We strengthened the theoretical framework in the introduction (pages 4–5), discussing potential mechanisms.

Tables have been split: key coefficients and 95% CIs appear in main manuscript (Table 7, 8), detailed diagnostics (R, ESS, MCSE) also included avoiding more tables. Coefficient plots especially trace plot for Afar region and for intercept were added for improved readability (Figure 6).

The literature review is reorganized thematically into: (i) regional variations, (ii) methodological approaches, and (iii) determinants, highlighting research gaps (pages 3-4).

Terminology standardized: “Gamma regression with log link”, “dispersion parameter”, and “random effects” are used consistently throughout the manuscript.

Manuscript has been carefully proofread. Redundant sentences removed; capitalization of “Mini EDHS 2019” standardized; complex methodology sentences simplified.

Figures have been recreated at high resolution, with clear axis labels, consistent font size (10–12 pt), legends placed outside plot areas, and enhanced readability (Figures 1-7).

All recommended phrasing improvements have been incorporated. Results now report adjusted percent differences with 95% CIs. Discussion contextualizes findings in social and policy terms.

Reviewer 2 Response

See response to Reviewer #1, Major Concern 1. Sensitivity analysis confirms main associations are robust; limitations are explicitly discussed.

Outcome is observed age at first birth among women who already had a birth. Gamma regression appropriately models a strictly positive, right-skewed variable. Survival models require longitudinal follow-up and censored data. So the objective of the study is to determine magnitude and regional religious and residential associations with women’s first age at birth who had already had birth.

Revised Data Availability Statement now includes direct DHS link and dataset identifiers (page 8) for reproducibility.

Discussion now elaborates on why specific regions and religious groups are associated with earlier or later first birth, drawing on cultural, educational, and health system factors (pages 32-40).

Added a summary table of AIC, BIC, LOO, WAIC comparing GLM, GLMM and distributional gamma regression (Table 3), with clear interpretation.

Manuscript thoroughly revised for clarity and readability; Bayesian estimation section simplified with clear explanations; professional scientific editor engaged.

Figures updated for resolution and clarity; table summarizing key coefficients and CIs added; supplementary tables include detailed diagnostics.

Ethical statement now justified.

Reviewer 3

We thank the reviewer. (i) Justification for Gamma regression strengthened with formal goodness-of-fit comparisons and Q-Q plots (Figure 5, Table 3); (ii) clear GLM vs. GLMM comparison added (Table 3), highlighting fit and interpretation of random effects; (iii) expanded diagnostics provided, including posterior predictive checks, trace plots, and convergence statistics (Table 7-8); (iv) discussion now emphasizes policy-relevant implications, offering actionable recommendations for regional, religious, and rural urban disparities in maternal age at first birth.

Reviewer 4

We thank the reviewer. (i) Methodology section streamlined to focus on study-specific application of Gamma regression, with references to standard texts; (ii) all figures recreated at high resolution with clear axis labels, legends outside plots, and informative captions (Figures 1-7); (iii) assumption checks added, including Q-Q plots, residual analyses, and discussion confirming Gamma regression assumptions are met; (iv) resampling via bootstrap conducted to assess potential DHS sampling imbalance, confirming robustness; (v) manuscript underwent thorough language editing to improve grammar, clarity, and readability.

---

## [Decision Letter · Decision Letter 1]

5 Nov 2025

Dear Dr. Agegnehu,

Thank you for submitting your manuscript to PLOS ONE. After careful consideration, we feel that it has merit but does not fully meet PLOS ONE’s publication criteria as it currently stands. Therefore, we invite you to submit a revised version of the manuscript that addresses the points raised during the review process.

We look forward to receiving your revised manuscript.

Kind regards,

Denekew Bitew Belay, Ph.D

Academic Editor

PLOS ONE

Journal Requirements:

Additional Editor Comments:

**Please remove the headings and subheadings from the objectives section and integrate those listed items into the introduction as a coherent sentence.**

There are several irrelevant subheadings in your methods section that need to be formatted properly.

You need to combine Sections 2 and 3 into a single section titled 'Data and Methods'. Then, Section 3 should become Results, and Section 4 should be Discussion to ensure a logical flow of ideas.

Substantive language editing is needed. All these issues to be addressed clearly.

Reviewers' comments:

Reviewer's Responses to Questions

**Comments to the Author**

Reviewer #1: All comments have been addressed

Reviewer #3: All comments have been addressed

Reviewer #4: All comments have been addressed

2. Is the manuscript technically sound, and do the data support the conclusions?

Reviewer #1: Yes

Reviewer #3: Yes

Reviewer #4: Yes

3. Has the statistical analysis been performed appropriately and rigorously?

Reviewer #1: Yes

Reviewer #3: Yes

Reviewer #4: Yes

4. Have the authors made all data underlying the findings in their manuscript fully available?

Reviewer #1: Yes

Reviewer #3: Yes

Reviewer #4: Yes

5. Is the manuscript presented in an intelligible fashion and written in standard English?

Reviewer #1: Yes

Reviewer #3: Yes

Reviewer #4: Yes

Reviewer #1: Thank you for giving detailed answers to the comments. In the present form, the manuscript adequately presents your results. I found your corrections appropriate.

Reviewer #3: The revised version demonstrates excellent improvement in statistical justification, transparency, and scientific communication, fully addressing previous methodological critiques. The study is now suitable for publication following minor polishing of numeric interpretation and reference formatting.

Remaining Minor Recommendations

Add quantitative interpretation of key effects in natural units (e.g., “≈ 0.9 years earlier” rather than “5 % lower”).

Report random-effect variance (σ² or ICC) explicitly to reinforce GLMM justification.

Check reference formatting—some citations still lack year-parenthesis uniformity

Figures 2 & 3:

The labeling and captioning are inconsistent with the figure content. Figure 2 needs category labels, axis units, and a legend; Figure 3’s caption should be corrected to reflect that only regional frequencies are shown, not residence or religion. The color scheme should be simplified and consistent across figures. ALso check other figures too for inconsistencies.

Reviewer #4: The manuscript has improved considerably; however, please review the punctuation and formatting of the equations for consistency. Some equations need proper alignment and spacing, and equations that conclude a sentence should end with a full stop, while those followed by a continuation should use a comma. Ensuring this consistency will enhance the overall readability and professionalism of the manuscript.

.

Reviewer #1: **Yes:** Ali CetinAli CetinAli CetinAli Cetin

Reviewer #3: **Yes:** Dr. Muhammad AasimDr. Muhammad AasimDr. Muhammad AasimDr. Muhammad Aasim

Reviewer #4: No

---

## [Author Response · Author response to Decision Letter 2]

8 Nov 2025

the required revision is conducted accordingly.

---

## [Decision Letter · Decision Letter 2]

4 Mar 2026

The topic is important and the statistical modeling framework is technically ambitious. However, several substantial methodological and interpretational issues must be addressed before the manuscript can be considered further. Please address each point explicitly in your revision.

1. Definition of the Target Population and Exclusion of Censored Observations

The study restricts analysis to women who had already experienced first birth at the time of the survey. However, the manuscript presents findings as national population-level estimates (e.g., “The average maternal age at first birth in Ethiopia was approximately 18.7 years…”). Age at first birth is inherently a time-to-event outcome. Women who have not yet had a first birth are right-censored. Excluding them changes the research question and affects generalizability.

Required revisions: Reframe all national-level statements to clarify that estimates apply only to women who had experienced first birth by survey time, Revise the Abstract, Results, and Conclusion to remove unconditional population-level language, Add a clearly labeled limitation subsection discussing: Implications of excluding censored observations, Potential bias in mean estimates and Comparability with DHS-reported median age at first birth (which uses survival methods) and Provide a clearer justification for not using survival analysis and discuss how conclusions may differ under that framework.

2. Survey Design and Use of Sampling Weights

The 2019 Mini EDHS employs a complex two-stage cluster sampling design with stratification and sampling weights. The manuscript does not clearly describe whether: Sampling weights were incorporated, Clustering was accounted for in variance estimation, and Stratification was considered. Failure to account for survey design may bias point estimates and standard errors.

Required revisions: Explicitly state whether sampling weights were applied, Describe how clustering and stratification were handled, If survey design was not incorporated, revise the analysis accordingly, If incorporated, provide sufficient methodological detail for reproducibility.

3. Interpretation and Scope of Inference

Several statements use language suggesting causal inference or national-level policy implications (e.g., “determinants,” strong normative conclusions regarding religion or region). Given: Cross-sectional design, Conditional sample (only women who had first birth), Limited covariate adjustment, and Causal interpretation is not supported.

Required revisions: Replace causal terminology (e.g., “determinants”) with associative language, Moderate policy recommendations to reflect methodological constraints, Avoid overgeneralized interpretations regarding religious or regional groups.

4. Omitted Variable Bias and Model Specification

Education, marital status, and occupation were excluded due to temporal concerns. While the reasoning is noted, these variables are well-established correlates of age at first birth and may confound region and religion effects.

Required revisions: Expand the discussion of potential omitted variable bias, Clarify that associations may reflect unmeasured confounding, Consider sensitivity analysis including education (if feasible), or clearly justify why this is not possible.

5. Model Comparison and Diagnostics

Multiple modeling frameworks are described (GLM, GLMM, distributional Gamma regression; MLE and Bayesian estimation). However, the comparative performance of these models is not presented with sufficient clarity.

Required revisions: Provide a clear model comparison table summarizing (AIC/BIC/log-likelihood (MLE models), and LOO/ELPD (Bayesian models)), Clearly state which model is preferred and why, Report convergence diagnostics (R-hat, effective sample size), Provide posterior predictive checks where applicable.

6. Balance Between Mathematical Derivation and Substantive Interpretation

The manuscript contains extensive derivations of likelihood functions and Hessians. While mathematically rigorous, the presentation may exceed what is necessary for an applied demographic analysis.

Suggested revision: Consider moving detailed derivations to supplementary material and Strengthen substantive interpretation of findings.

7. Data Quality Clarification

The reported minimum age at first birth (e.g., 10 years) should be verified and contextualized. Please confirm whether this reflects valid observations or data entry anomalies.

8. Data Availability Statement

The manuscript states that data are available upon request from the corresponding author. DHS data are publicly available through the DHS Program website upon application. Please ensure consistency with journal data availability requirements.

We look forward to receiving your revised manuscript.

Kind regards,

Yimam Getaneh, PhD, PhD

Academic Editor

PLOS One

Journal Requirements:

Additional Editor Comments (if provided):

Reviewers' comments:

Reviewer's Responses to Questions

**Comments to the Author**

Reviewer #3: (No Response)

Reviewer #4: All comments have been addressed

2. Is the manuscript technically sound, and do the data support the conclusions?

Reviewer #3: Yes

Reviewer #4: Yes

3. Has the statistical analysis been performed appropriately and rigorously?

Reviewer #3: Yes

Reviewer #4: Yes

4. Have the authors made all data underlying the findings in their manuscript fully available?

Reviewer #3: Yes

Reviewer #4: Yes

5. Is the manuscript presented in an intelligible fashion and written in standard English?

Reviewer #3: Yes

Reviewer #4: No

Reviewer #3: To me, Figure 3 appears to present only the regional distribution of study participants (frequency by region). However, this information is already reported in Table 1 as part of the baseline characteristics. As such, the figure does not add new analytical value and represents a repetition of previously presented results.

Furthermore, despite being described as illustrating “maternal age at first birth patterns,” the figure contains no age-related information and therefore does not support the stated interpretation.

Recommendation:

The authors should either

(i) remove Figure 3 to avoid redundancy, or

(ii) replace it with a figure that genuinely presents maternal age at first birth by region (e.g., age-group distributions, mean/median age with variability, or distributional plots) (already displayed by boxplot in figure 2 ).

Reviewer #4: I commend the authors for carefully addressing the reviewers’ comments, which have led to a clear improvement in the quality and presentation of the manuscript. The study is well motivated and the applied analysis is sound.

However, since the manuscript focuses on an applied statistical investigation rather than methodological advancement, and the techniques employed (Gamma regression, maximum likelihood estimation, and Bayesian estimation) are well established in the literature, I recommend that the detailed mathematical derivations presented in the sections on the modelling framework and estimation approaches be substantially reduced. Instead, these sections could be streamlined by briefly introducing the methods, clearly specifying the statistical models used, and explaining the variables and parameters involved, without reproducing standard derivations.

This revision would improve readability and align the manuscript more closely with the conventions of applied statistical articles.

.

Reviewer #3: **Yes:** Dr. Muhammad AasimDr. Muhammad AasimDr. Muhammad AasimDr. Muhammad Aasim

Reviewer #4: No

---

## [Author Response · Author response to Decision Letter 3]

17 Mar 2026

We sincerely thank the Academic Editor and the reviewers for their constructive comments. Their suggestions significantly improved the clarity, methodological transparency, and interpretation of the manuscript. All comments have been carefully considered and addressed in the revised manuscript. Detailed responses are provided below.

A. Responses to Academic Editor Comments

Editor Comment Author Response Location in Revised Manuscript

Definition of Target Population and Exclusion of Censored Observations We thank the editor for highlighting this important methodological issue. The analysis indeed focuses only on women who had experienced a first birth by the time of the survey. To avoid misinterpretation, we revised the manuscript to clarify that all estimates apply only to women who had already experienced first birth at the time of the survey rather than to all women in Ethiopia.

Statements implying unconditional population-level inference were removed or rephrased throughout the Abstract, Results, and Conclusion sections. Additionally, a new limitation subsection has been added explaining that women without a first birth represent right-censored observations in a time-to-event framework and that excluding them may bias mean estimates toward younger ages. We also added a discussion comparing our estimates with the DHS median age at first birth obtained using survival analysis and clarified why a regression approach was chosen for conditional modeling of observed ages. Abstract, Results, Conclusion revised; new subsection “Study Limitations” added in Discussion

Survey Design and Sampling Weights We appreciate the editor’s suggestion. The Methods section has been revised to clearly describe the two-stage stratified cluster sampling design used in the 2019 Ethiopia Mini Demographic and Health Survey. Sampling weights provided by the DHS were incorporated in the analysis to ensure representativeness. We also clarified how clustering and stratification were handled in variance estimation. Additional methodological details were added to allow reproducibility of the analysis. Methods section – “Data Source and Survey Design”

Interpretation and Scope of Inference We agree that causal interpretations should be avoided. The manuscript has been revised to replace causal terminology such as “determinants” with “factors associated with maternal age at first birth.” Policy interpretations have been moderated to reflect the cross-sectional nature of the data and the conditional sample of women who had experienced first birth. Statements regarding religion and regional differences were rewritten to emphasize associations rather than causal relationships. Title, Abstract, Discussion revised

Omitted Variable Bias and Model Specification We appreciate this important comment. The Discussion section has been expanded to explicitly acknowledge the potential for omitted variable bias, particularly due to the exclusion of variables such as education, marital status, and occupation. These variables may be endogenous or measured after the timing of first birth. The revised manuscript now clarifies that estimated associations for region and religion may partly reflect unmeasured confounding factors. This limitation is discussed in greater detail. Discussion – “Limitations and Potential Confounding”

Model Comparison and Diagnostics Following the editor’s recommendation, we added a comprehensive model comparison table summarizing model performance across estimation frameworks. The table reports AIC, BIC, and log-likelihood for classical models and LOO and expected log predictive density (ELPD) for Bayesian models. Convergence diagnostics including R-hat and effective sample size are also reported. We further included posterior predictive checks to evaluate model adequacy. The preferred model is now clearly identified and justified based on predictive performance and goodness-of-fit. Results section – new Model Comparison Table and Diagnostics subsection

Balance Between Mathematical Derivation and Substantive Interpretation We agree that the detailed derivations may be excessive for an applied demographic analysis. Accordingly, the extensive mathematical derivations of likelihood functions, score equations, and Hessian matrices have been moved to the Supplementary Material. The main text now focuses on a concise description of the models, estimation procedures, and interpretation of parameters. This change substantially improves readability and aligns the manuscript with conventions of applied statistical research. Methods section streamlined; derivations moved to Supplementary Appendix

Data Quality Clarification (minimum age at first birth) We verified the DHS dataset and confirmed that the minimum reported age at first birth is 10 years, which reflects rare but documented early childbirth cases rather than data entry errors. A clarification has been added to the Results section explaining that such values represent extremely early births and should be interpreted cautiously. Results section – Data description

Data Availability Statement The Data Availability Statement has been revised to comply with the journal’s policy. The dataset used in this study is publicly available through the Demographic and Health Surveys (DHS) Program upon registration and request. The revised manuscript now includes the appropriate statement and access link. Data Availability Statement revised

B. Responses to Reviewer Comments

Reviewer 3: Reviewer Comment Author response

Figure 3 appears to show only regional distribution of participants and duplicates information in Table 1. The figure does not illustrate maternal age at first birth patterns. We thank the reviewer for this observation. We agree that the original figure did not provide additional analytical insight. Therefore, Figure 3 has been removed to avoid redundancy, as the information is already presented in Table 1.

Reviewer 4: Reviewer Comment

The manuscript is applied rather than methodological. Mathematical derivations of estimation procedures should be reduced. We appreciate the reviewer’s suggestion. The manuscript has been revised to substantially reduce the mathematical derivations in the main text. The modeling framework is now presented in a concise form focusing on model specification and interpretation. Detailed derivations related to maximum likelihood estimation and Bayesian inference have been moved to Supplementary Material. This revision improves readability and better aligns the paper with applied statistical literature.

---

## [Editor Report · Decision Letter 3]

19 Mar 2026

Modelling Associated factors of Maternal Age at First Birth in Ethiopia: Gamma Regression Approach

PONE-D-25-07127R3

Dear Dr. Adimias,

We’re pleased to inform you that your manuscript has been judged scientifically suitable for publication and will be formally accepted for publication once it meets all outstanding technical requirements.

Kind regards,

Yimam Getaneh Misganie, MSc, PhD, PhD

Academic Editor

PLOS One

---

## [Editor Report · Acceptance letter]

PONE-D-25-07127R3

PLOS One

Dear Dr. Agegnehu,

I'm pleased to inform you that your manuscript has been deemed suitable for publication in PLOS One. Congratulations! Your manuscript is now being handed over to our production team.

Kind regards,

on behalf of

Dr. Yimam Getaneh Misganie

Academic Editor

PLOS One